# Gephyrin-Lacking PV Synapses on Neocortical Pyramidal Neurons

**DOI:** 10.3390/ijms221810032

**Published:** 2021-09-17

**Authors:** Dika A. Kuljis, Kristina D. Micheva, Ajit Ray, Waja Wegner, Ryan Bowman, Daniel V. Madison, Katrin I. Willig, Alison L. Barth

**Affiliations:** 1Center for the Neural Basis of Cognition, Department of Biological Sciences, Carnegie Mellon University, Pittsburgh, PA 15213, USA; dika.kuljis@gmail.com (D.A.K.); ajitr@andrew.cmu.edu (A.R.); ryan.bowman1802@gmail.com (R.B.); 2Department of Molecular and Cellular Physiology, Stanford University, Palo Alto, CA 94304, USA; Kristina.Micheva@stanford.edu (K.D.M.); madison@stanford.edu (D.V.M.); 3Optical Nanoscopy in Neuroscience, Center for Nanoscale Microscopy and Molecular Physiology of the Brain, University Medical Center Göttingen, 37075 Göttingen, Germany; waja.wegner@t-online.de (W.W.); kwillig@em.mpg.de (K.I.W.); 4Max Planck Institute of Experimental Medicine, 37075 Göttingen, Germany

**Keywords:** synapses, parvalbumin, gephyrin, intrabody, GABA_A_ receptors, synaptophysin, optogenetics, fluorescent protein sensors, light microscopy, correlative fluorescence and electron microscopy

## Abstract

Gephyrin has long been thought of as a master regulator for inhibitory synapses, acting as a scaffold to organize γ-aminobutyric acid type A receptors (GABA_A_Rs) at the post-synaptic density. Accordingly, gephyrin immunostaining has been used as an indicator of inhibitory synapses; despite this, the pan-synaptic localization of gephyrin to specific classes of inhibitory synapses has not been demonstrated. Genetically encoded fibronectin intrabodies generated with mRNA display (FingRs) against gephyrin (Gephyrin.FingR) reliably label endogenous gephyrin, and can be tagged with fluorophores for comprehensive synaptic quantitation and monitoring. Here we investigated input- and target-specific localization of gephyrin at a defined class of inhibitory synapse, using Gephyrin.FingR proteins tagged with EGFP in brain tissue from transgenic mice. Parvalbumin-expressing (PV) neuron presynaptic boutons labeled using Cre- dependent synaptophysin-tdTomato were aligned with postsynaptic Gephyrin.FingR puncta. We discovered that more than one-third of PV boutons adjacent to neocortical pyramidal (Pyr) cell somas lack postsynaptic gephyrin labeling. This finding was confirmed using correlative fluorescence and electron microscopy. Our findings suggest some inhibitory synapses may lack gephyrin. Gephyrin-lacking synapses may play an important role in dynamically regulating cell activity under different physiological conditions.

## 1. Introduction

Genetically-encoded, fluorescent synapse markers are advantageous for high-throughput, quantitative analysis of cell-type and input-specific synaptic changes during development, learning, and disease [1,2,3,4,5,6,7,8,9]. Methodologies for synapse identification have typically focused on electron microscopy (EM) analysis using volumetric reconstruction, a technique that is computationally intensive and limited in its ability to label multiple molecular markers helpful for cell-type specific classification. Molecular genetic methods for fluorescence-based synapse quantitation are useful as they enable cell-type specific analysis, are compatible with other methods for protein or input-specific labeling, and are easily adopted by diverse communities of neuroscience researchers because they do not require specialized instrumentation for data collection and analysis. Although fluorescence microscopy lacks the resolution of EM, synapse detection can be facilitated using multiple markers for synapse identification to reduce error rates [10]. Sparse and cell-type specific expression of synapse-labeling constructs enables precise attribution of synapses to specific classes of presynaptic inputs, critical for understanding how networks are assembled and process information [11,12,13,14,15,16,17,18].

Previously we have developed methodologies for pan-synaptic labeling and input-specific analysis in fluorescently labeled tissue using a post-synaptic marker (FAPpost) [3] derived from a truncated version of neuroligin-1 (NL-1) [8]. Elimination of the neurexin-associated extracellular domain in FAPpost resulted in labeling of both excitatory and inhibitory synapses [3], consistent with the conserved PSD-95 and gephyrin binding domains in the cytoplasmic region of NL-1 [19]. Sparse neuronal expression of this construct in defined cell-types enabled the visualization of synapse distribution across the entire soma to dendritic arbor of an individual neuron, where multiple types of presynaptic inputs could be aligned with postsynaptic puncta. However, postsynaptic markers that selectively label either excitatory or inhibitory inputs in combination with cell-type specific presynaptic labeling, may be useful for more focused investigations, such as defining cortical inhibitory synapses that arise from a diverse array of presynaptic GABAergic neurons [20].

Gephyrin is widely thought to be structurally important at inhibitory synapses [21,22,23,24,25,26,27,28,29] and has been used as an indicator for GABAergic synapses in quantitative analyses [1,6,30,31,32]. Genetically-encoded fibronectin intrabodies generated with mRNA display against gephyrin (Gephyrin.FingR) have been used for the fluorescent labeling of gephyrin in living cells [1,33,34,35], where levels of FingR fluorescence are thought to be proportional to the number of gephyrin multimers at synapses. Consistent with prior studies [33,34,35], we found that EGFP-tagged Gephyrin.FingR showed strong colocalization with immunolabeled gephyrin in virally-transduced neurons. 

To determine whether Gephyrin.FingR could be employed to track GABAergic synapses of a given input class, we focused on its association with parvalbumin-expressing (PV) neuron inputs to pyramidal (Pyr) neurons. PV cells are a major source of inhibition onto cortical Pyr neurons, targeting both their soma and dendrites for fast feedback inhibition that shapes cortical firing [3,36,37,38]. Other studies have indicated that these synapses can be plastic in the face of alterations in sensory input and learning [1,6,33,39,40,41], suggesting that quantitative analysis and in vivo monitoring of these synapses might be highly informative. To identify PV synapses onto cortical Pyr neurons, we labeled PV boutons with a Cre-dependent tdTomato-tagged synaptophysin [42] in PV-Cre transgenic mice and digitally aligned these boutons with Gephyrin.FingR puncta on Pyr neurons in layer 2/3 (L2/3) and L5 in mouse somatosensory (barrel) cortex. Surprisingly, our analysis revealed that postsynaptic gephyrin was absent from more than 30% of presynaptic PV basket boutons onto Pyr neuron somas, findings that were corroborated using fluorescence-correlated immuno-EM. PV synapses onto PV inhibitory neurons also showed a high level of putative synaptic contacts that lacked Gephyrin.FingR labeling. These data indicate that gephyrin may not be a required constituent of PV inhibitory synapses and suggest that gephyrin-lacking PV synapses may play a distinct functional role.

## 2. Results

### 2.1. Gephyrin as an Inhibitory Synapse Marker in Cortical Pyramidal Neurons

Gephyrin is an attractive molecular candidate for pan-inhibitory synaptic labeling, due to the presumed ubiquity of its expression at inhibitory synapses [21,22,23,24,25,26,27,28,29]. Transgenic overexpression of fluorescent proteins fused to gephyrin and Gephryin.FingRs have previously been used to label cortical inhibitory synapses, as well as track their plasticity on dendrites [1,6,30,34]. 

To label gephyrin in neurons, we used a virally-transduced construct consisting of an intrabody-like protein Gephyrin.FingR, that binds to endogenous gephyrin, in combination with a transcriptional regulation system consisting of the DNA-binding zinc finger domain and the transcriptional repressor domain KRAB-A [33] (Figure 1A). By design, Gephyrin.FingR expression is controlled by negative feedback regulation so that once endogenous gephyrin binding sites are occupied, unbound Gephyrin.FingR is trafficked to the nucleus due to a nuclear localization sequence that is part of the zinc finger domain. Binding of the repressor KRAB-A to the promoter via the zinc finger inhibits further transcription and expression of Gephyrin.FingR [33]. By this mechanism, excess background fluorescence in the dendrites is reduced, and nuclear fluorescence is an indicator of saturating levels of Gephyrin.FingR at synapses [33].

We generated recombinant adeno-associated viral (AAV) particles encoding fusion proteins of Gephyrin.FingR, EGFP and the regulation system under control of the human Synapsin promoter (hSyn) for robust expression in mammalian neurons (AAV1/2-ZFN-hSyn-Gephyrin.FingR-EGFP-IL2RGTC-KRAB-A). The virus was stereotaxically delivered into the primary somatosensory (barrel) cortex of juvenile mice (Figure 1B,C). One to three weeks later, fixed tissue was prepared and Gephyrin.FingR-expressing L2/3 and L5 Pyr neurons were selected for volumetric confocal imaging and computer-assisted quantitative analysis (Figure 2). Due to the high density of transduced neurons, our analysis focused on the soma of Pyr neurons, where Gephyrin.FingR puncta could be more easily attributed to a single cell (Figure 2A–C). The density of somatic Gephyrin.FingR puncta was heterogeneous, varying more than 10-fold across neurons. On average, L5 Pyr neuron somas showed higher overall numbers of Gephyrin.FingR puncta than L2/3 Pyr neuron somas, possibly because of their larger soma size (gephyrin count per soma L2/3 = 76 ± 49 vs. L5 = 132 ± 117; Figure 2D). Indeed, the density of somatic Gephyrin.FingR puncta was remarkably similar across lamina (L2/3 = 0.12 ± 0.07 per µm^2^ vs. L5 = 0.13 ± 0.09 per µm^2^; Figure 2E). 

To determine whether heterogeneity in the number or intensity of Gephyrin.FingR puncta might be attributed to variable levels of construct expression in different cells, we compared puncta measurements per cell with nuclear fluorescence. We observed no correlation between the intensity or density of Gephyrin.FingR puncta and nuclear fluorescence (Figure 2F,G), suggesting that this variability may faithfully reflect differences in inhibition across Pyr neurons.

Importantly, a comparison between gephyrin immunofluorescence (IF) and Gephyrin.FingR labeling in neocortical neurons showed a strong correlation between the two labeling modalities (Figure 3). Overall, we conclude that Gephyrin.FingR localization and intensity reflects levels of endogenous gephyrin protein in cortical neurons, and that the number of gephyrin-labeled inhibitory synapses per cell may vary considerably across neocortical Pyr neurons.

### 2.2. Gephyrin Colocalization at PV Synapses onto Pyr Neurons

Somatic inhibition of neocortical Pyr neurons is dominated by PV basket cells [3,20], and fast feedback inhibition from PV cells can potently suppress recurrent activity within the cortical circuit [43]. Furthermore, inhibition from PV neurons can be regulated during sensory or behavioral experience [44,45], and anatomical methods to quantitate PV inputs might be useful in many different experimental paradigms. We crossed mice carrying a Cre-dependent tdTomato-tagged synaptophysin (Syn-tdTomato; Ai34 transgenic mice) [42] to PV-Cre [46] animals to label PV axon terminals with Syn-tdTomato (PV-Syn). Using Gephyrin.FingR labeling in cortical Pyr neurons, we used volumetric confocal imaging to quantitate the frequency at which gephyrin puncta were adjacent to PV terminals. PV-Syn boutons could be easily detected around Pyr cell somas and could be aligned with gephyrin puncta in single imaging planes (Figure 4A–F). In general, presynaptic PV boutons appeared much larger than postsynaptic gephyrin puncta, consistent with postsynaptic specializations for GABAergic neurotransmission being more spatially constrained than the entire presynaptic bouton. 

Because gephyrin is thought to be a critical component of the inhibitory synapse required for the clustering and stabilization of GABA_A_Rs, we were surprised to sometimes observe PV boutons onto Pyr soma without underlying postsynaptic Gephyrin.FingR puncta (Figure 4A–F, asterisk). Serial section analysis of confocal image stacks revealed that the absence of gephyrin puncta could not simply be attributed to the larger size of PV boutons and image selection, as analysis of multiple planes did not reveal gephyrin fluorescence. Thus, a fraction of PV inputs may lack the trimeric gephyrin targeted by the Gephyrin.FingR intrabody. 

Additionally, not all Gephyrin.FingR puncta were associated with presynaptic PV boutons (Figure 4C–F), suggesting that they might be linked to another somatically-innervating inhibitory neuron subtype, such as cholecystokinin (CCK) basket cells [20].

### 2.3. Automated Alignment and Quantitation

The frequency of Gephyrin.FingR-lacking PV boutons was quantitated using an object-based image analysis approach using Imaris software (v8.4 with FilamentTracer, Bitplane AG, Zurich, Switzerland; Figure 5). Based on resolution limits of conventional confocal microscopy, we expected some overlap or adjacency between pre- and postsynaptic structures for any given synapse, and thus used a distance threshold between the edges of 3D digital renderings of PV boutons and gephyrin puncta to assess alignment. Boutons adjacent to or touching gephyrin puncta were classified as aligned with gephyrin (Figure 5F), and those beyond the threshold were classified as not aligned (Figure 5G,H). The same approach was used to classify all gephyrin puncta (Figure 5I) as aligned with PV boutons (PV synapses; Figure 5J) or not aligned (non-PV synapses; Figure 5K,L). 

The frequency of PV boutons aligned with postsynaptic gephyrin puncta was heterogenous across individual Pyr neurons (Figure 6). PV boutons on some somas nearly always had corresponding gephyrin (Figure 6A,B), and for other Pyr neurons we found that most somatic PV boutons lacked gephyrin (Figure 6C). Similar to the absolute numbers of Gephyrin.FingR puncta across individual L2/3 and L5 Pyr neurons, we also saw a similar layer-specific difference in the number of PV boutons associated with these cells. On average, there were ~100 PV boutons surrounding L2/3 Pyr neuron somas, and ~125 surrounding L5 Pyr neurons, a difference that was significant between layers (Figure 6D). Overall PV bouton density was not different between layers, (L2/3 = 0.14 ± 0.02 per µm^2^ vs. L5 = 0.15 ± 0.02 per µm^2^; Figure 5E). 

PV boutons without gephyrin signal showed a similar frequency across layers. For both L2/3 and L5 Pyr neurons, on average ~60% of individual PV boutons lacked associated gephyrin puncta (Figure 6F). A positive correlation between gephyrin puncta and PV bouton numbers per cell might suggest these synaptic markers were related if not spatially coupled; however, across all Pyr neurons examined, gephyrin and PV bouton density were not correlated with each other (R^2^ = 0.01, *p* = 0.6; data not shown). 

The fraction of gephyrin puncta without a corresponding PV input varied widely across Pyr cells of both L2/3 and L5, ranging from 2 to 50% (Figure 6G). On average, nearly a quarter of somatic gephyrin puncta were not aligned with PV boutons. These data suggest that individual Pyr neurons in the neocortex do not receive a stereotyped and fixed amount of inhibition from molecularly defined GABAergic cell types; rather, some cells are dominated by PV inhibition and others show a balance between PV and other types of inhibition.

### 2.4. Conjugate Immunofluorescence and Electron Array Tomography Confirm Gephyrin-Lacking Inhibitory Synapses

Inhibitory synapses were inferred based upon the alignment of fluorescently labeled pre- and postsynaptic markers, but for gephyrin-lacking PV contacts, it was possible that PV boutons were adjacent to the soma of interest but synapsing with a different postsynaptic target. EM enables visualization of ultrastructural features such as presynaptic vesicles, presynaptic mitochondria, the synaptic cleft, and the postsynaptic density to act as indicators of an inhibitory synapse. We used antibodies against gephyrin and GABA to identify inhibitory synapses onto presumed glutamatergic neurons from tissue specimens prepared from superficial layers of mouse barrel cortex (Figure 7). 

Serial section EM analysis indicated there was a subset of contacts where either no gephyrin could be detected, or it was very low and visible only in a single section through the synapse (Figure 7B). This was prominent at the soma, where more than 35% of GABA inputs showed no gephyrin-IF, and a similar percentage showed only a single thin section with detectable gephyrin-IF. On dendrites, more than 85% of GABAergic inputs exhibited gephyrin-IF in one or more consecutive sections. Because PV interneurons are the most abundant type of inhibitory neurons in the neocortex [47], and a substantial portion of their output synapses are concentrated on the soma [20], it is highly likely that at least some of the gephyrin-negative inhibitory synapses detected from EM images are from PV neurons.

For the above analysis, the tissue was embedded in highly cross-linked acrylate-based resin to ensure adequate ultrastructure for correlative immunofluorescence-SEM analysis. To verify that the results were not influenced by reduced antigenicity due to tissue processing, we also estimated gephyrin content at inhibitory synapses in tissue embedded in a different resin, LR White, which better preserves antigenicity to the detriment of ultrastructure. These samples also provided the opportunity to specifically identify PV+ inhibitory synapses using PV immunofluorescence. Synapses were detected using immunofluorescent array tomography, which provides a higher resolution (200 nm lateral resolution and 70 nm axial resolution) compared to confocal microscopy of thick sections, and is sufficient to identify most neocortical synapses [48]. In these datasets from the barrel cortex of two adult mice, synapses were identified by the concomitant presence of the presynaptic markers GAD, VGAT and synapsin on at least two serial sections, and an immediately adjacent postsynaptic marker GABA_A_Rα1 on one section. One of the mice was a YFP-H mouse that expressed YFP in a fraction of L5 Pyr neurons, which enabled the analysis of synapses onto Pyr cell bodies (Figure 6F). The results were also confirmed in a wild-type mouse, where somatic synapses onto layer 5 PV-negative neurons were analyzed, and these likely included pyramidal neurons and PV-negative interneurons. In both mice, there were PV+ inhibitory synapses with undetectable gephyrin labeling, even though other nearby inhibitory synapses exhibited robust gephyrin immunolabel on multiple consecutive sections (up to 6 or 7 in some cases). PV synapses without gephyrin immunolabel were more commonly observed at somatic locations (Figure 7G). In the YFP-H mouse, 21% of PV+ synapses on pyramidal neuron cell bodies (12 out of 56) had no detectable gephyrin-IF, and in the wild-type mouse 33% of PV+ synapses on PV-negative cell bodies (9 out of 27) had no gephyrin-IF. In comparison, only 11% (9 out of 81 from the 2 mice) of PV synapses on dendrites and spines lacked gephyrin-IF. This analysis provides further support for the existence of PV somatic synapses with little or no gephyrin content.

### 2.5. Quantitative Analysis of Inhibitory Synapses on Pyr Neurons

The size of an axon bouton is likely to be related to its presynaptic release properties and thus the strength of the synapse [49]. Digital fluorescence analysis enabled us to compare the sizes of soma-associated PV boutons and gephyrin puncta for L2/3 and L5 Pyr neurons. We identified 2623 somatic PV boutons (1167 in L2/3 and 1456 in L5), and subdivided them according to whether they were associated with gephyrin or not (Figure 8). PV boutons and Gephyrin.FingR puncta exhibited a wide range of volumes with a non-normal distribution. Because of this, we compared values using a cumulative distribution and median values for each cell in the dataset (Figure 8A–D). Both large and small boutons aligned with postsynaptic gephyrin (Figure 8A,B) and PV boutons were similar in volume across layers (median L2/3 = 0.61 ± 0.08 µm^3^, L5 = 0.66 ± 0.22 µm^3^, *p* = 0.9). 

Across individual Pyr neurons, median PV bouton volume was nearly 2-fold larger for Gephyrin.FingR-aligned boutons, although this difference varied substantially across cells (1.1 to 4.6-fold-difference; Figure 8A,B). Additionally, aligned boutons had more intense PV-Syn fluorescence (+13 ± 8%; *p* < 0.001; Appendix A). These data suggest the absence of gephyrin might be correlated with functional properties of the synapses.

Synaptic gephyrin is a predictor of inhibitory synaptic strength [6], as it directly associates with GABA_A_Rs [50,51,52,53]. We used our anatomical data to evaluate whether PV bouton-aligned somatic gephyrin puncta were different than isolated somatic gephyrin puncta that likely belong to non-PV synapses, in particular CCK synapses, the other most abundant type of Pyr somatic innervation [54,55,56]. We identified 1952 somatic gephyrin puncta (935 in L2/3 and 1017 in L5) and subdivided them according to whether they were associated with presynaptic PV boutons (Figure 8C,D). The average PV-aligned gephyrin puncta volume was similar to non-PV gephyrin puncta in both L2/3 and L5 (Figure 8C,D), suggesting that gephyrin-containing somatic synapses are similar in size across input types. Irrespective of whether gephyrin could be aligned with presynaptic PV boutons, gephyrin puncta volumes in L5 were not significantly different across layers (L2/3 = 0.050 ± 0.022 µm^3^, L5 = 0.076 ± 0.052 µm^3^, *p* = 0.1; Appendix A). 

Previous studies have compared functional strength of L2/3 Pyr neuron inhibition across inhibitory cell-types used Cre-dependent channelrhodopsin (ChR2; Ai32 line) to stimulate synchronous GABA release from boutons of a particular class [36,57]. To relate fluorescent PV synapse markers to functional measurements, we examined the total volume of all somatic PV boutons or gephyrin puncta. Consistent with PV neurons being the most potent inhibitors of L2/3 Pyr neurons, the total volume of PV-aligned gephyrin puncta was significantly greater than the total volume of non-PV gephyrin puncta (PV = 4.1± 3.0 µm^3^ vs non-PV = 0.9 ± 0.5 µm^3^, *p* = 0.002; Appendix A). A similar pattern was observed for L5 Pyr neurons (PV = 9.1 ± 8.2 µm^3^ vs. non-PV = 1.0 ± 0.8 µm^3^, *p* = 0.01; Appendix A). Additionally, we determined that the summed volume of all PV boutons on individual L5 Pyr neurons was 50% greater than for L2/3 Pyr neurons (L2/3 = 71 ± 10 µm^3^ vs. L5 = 109 ± 35 µm^3^, *p* = 0.001; Figure 8E). Although there was a trend for the summed volume of gephyrin puncta aligned with PV boutons to also be larger in L5, this difference was not statistically significant (L2/3 = 4.1 ±3.0 µm^3^ vs. L5 = 9.1 ± 8.2 µm^3^, *p* = 0.2; Figure 8F). 

### 2.6. PV-Mediated Inhibition Is Functionally Stronger for L5 Pyr Neurons

Our data suggest that L5 Pyr neurons might receive stronger PV inhibition than L2/3 Pyr neurons, since they showed a greater overall number of soma-associated PV boutons, and a greater total volume of PV boutons (Figure 6D and Figure 8E). To determine whether these morphological differences are functionally correlated with larger inhibitory currents from PV neurons, we used PV-Cre x Ai32 (Cre-dependent ChR2) double transgenic mice for electrophysiological analysis of PV inhibitory postsynaptic currents (PV-IPSCs) across layers. Consistent with somatic synaptic fluorescence morphological measurements, PV-IPSC amplitude was approximately 80% larger for L5 Pyr neurons compared to L2/3 (L2/3 = 368 ± 183 pA vs L5 = 662 ± 267 pA; *p* = 0.01; Figure 9A,B). In a subset of experiments, we confirmed that light-evoked currents were inhibitory, as responses were abolished by the application of the GABA_A_R antagonist picrotoxin (Figure 9C). These results establish that PV terminals onto L2/3 and L5 Pyr neurons are inhibitory in nature, an important control since a minority of PV cells are glutamatergic [58,59]. Although the synaptic targets of excitatory PV neurons have not been established, these data indicate that they are not likely to synapse onto L2/3 or L5 Pyr neurons.

Together with morphological measurements of PV boutons and gephyrin puncta, these data suggest that PV input onto L5 Pyr neurons may be a more potent regulator of Pyr activity than in L2/3.

### 2.7. PV Synapses onto PV Neurons Also Lack Postsynaptic Gephyrin

The apparent absence of gephyrin at PV contacts onto Pyr neurons may be a property of Pyr neurons, or it may be observed at PV contacts onto other target cells. Indeed, both pre- and postsynaptic cell identity are likely to contribute to distinct structural and functional properties of GABAergic synapses. To investigate whether PV inputs onto other target neurons show the same degree of gephyrin colocalization, we investigated PV inputs onto other PV neurons, as it has been well-established that PV neurons robustly inhibit each other through both chemical and electrical synapses [11,36,60,61,62,63,64,65,66]. 

PV neurons across the cortical column were identified using faint cytoplasmic PV-Syn fluorescence labeling of PV neurons. A subset of PV neurons was transduced with Gephyrin.FingR AAV, and presynaptic PV inputs onto these target cells could be detected and evaluated for gephyrin colocalization (Figure 10). The range of Gephyrin.FingR puncta counts and somatic density for individual PV neurons was highly heterogeneous (~20-fold; Figure 9). Levels of PV innervation onto PV target neurons were less variable than Gephyrin.FingR puncta counts, with PV neuron somas exhibiting only a ~3-fold difference in PV bouton count and density across cells (Figure 10M,N). Some PV neurons showed dense PV bouton innervation at the soma but few gephyrin puncta, so that the fraction of PV boutons aligned with gephyrin was very low (<5% of the total PV bouton; Figure 10A–E, O). Other PV neurons showed dense somatic PV boutons, where most PV boutons had underlying gephyrin (>80% of PV boutons; Figure 10F–J,O). On average, we observed that the majority of the PV boutons onto PV neuron somas were not aligned with postsynaptic gephyrin (Figure 10O), similar to Pyr neurons (PV = 75 ± 21% vs. Pyr = 61 ± 21%, *p* = 0.08). 

PV neurons also receive inhibitory input from other cortical inhibitory neurons, most notably vasoactive intestinal peptide (VIP), CCK, and neurogliaform cells [11,66]. For a small number of cells, ~90% of somatic puncta could be aligned with PV boutons, but on average, ~40% of gephyrin puncta could not be linked to PV inputs, suggesting that they belonged to a separate input class (Figure 10O). This finding is consistent with previous reports indicating VIP and CCK synapses are the predominant source of somatic inhibition for most PV neurons [66,67,68], although our results also indicate a subset of PV neurons preferentially receive somatic PV input. Overall, a larger proportion of gephyrin puncta on PV neuron somas was not associated with presynaptic PV compared to Pyr neuron somas (PV = 44 ± 23% vs. Pyr = 26 ± 18% of gephyrin puncta were not associated with PV boutons, *p* = 0.03), consistent with the greater diversity of somatic inputs for this cell-type.

## 3. Discussion

Quantitative synapse analysis can be a useful method to assess circuit structure and plasticity during development, experience, and disease. Inhibitory synapses may be particularly plastic, where pathway-specific changes in inhibition have been detected in multiple brain areas across many different conditions [69,70]. Towards these ends, identification of a reliable pan-synaptic marker for inhibitory synapses could have many applications for neuroscientists. Although gephyrin is an attractive candidate molecule for inhibitory synaptic labeling, our data indicate that it may substantially underestimate synaptic contacts, particularly from PV neurons. Both genetically encoded fluorescent synaptic markers and immunofluorescence-correlated light and EM indicate gephyrin is absent from one-third or more of PV inhibitory somatic synapses on neocortical Pyr and PV neurons. Other postsynaptic markers, such as neuroligin-based tags (FAPpost and mGRASP) [3,8] may be better suited to the comprehensive detection of inhibitory inputs. Alternatively, the diversity of molecules at GABAergic synapses may preclude the development of a truly pan-synaptic marker for inhibitory synapses. 

### 3.1. Universal Marker for Inhibitory Synapses

The discovery and characterization of PSD-95 as a scaffolding protein for excitatory postsynaptic densities has allowed researchers to tag and quantitate glutamatergic synapses in a variety of different contexts and experiments [71,72,73,74]. Finding a molecular counterpart that universally labels inhibitory synapses has proved to be more complicated. Pentameric GABA_A_Rs can be assembled from 19 different subunits of eight different subunit classes in a staggering number of combinations [75], and the composition of GABA_A_Rs can vary greatly across cell type, development, subcellular location, and input [76]. Gephyrin colocalizes very strongly with several ubiquitous GABA subunits, including α1, α2, α3, and γ2 [53]. Because gephyrin directly interacts with inhibitory synaptic molecules, such as GABA_A_R subunits α1-3, β2-3 [23,26,77,78,79,80], as well as neuroligin-2 (NL2) and the adaptor protein collybistin [81,82,83], it has become widely accepted that gephyrin can be a general indicator for GABAergic synapses [27,28].

However, multiple lines of evidence indicate gephyrin is not required for GABAergic synapse function. In gephyrin knock-out mice, the density of GABA_A_R α1 and α5-containing synapses in the spinal cord are comparable to wild-type animals [84]. In addition, α2- and γ2 -GABA_A_Rs and miniature IPSC frequency are normal in hippocampal cultures from the gephyrin knock-out [85]. Gephyrin-independent clustering of GABA_A_Rs has been observed in other brain areas, including α1-containing GABA_A_Rs on Purkinje cell somas [86,87,88,89], perisomatic α1-containing GABA_A_Rs on hippocampal Pyr cells [90,91,92], and α2/α3/α5/β3-containing GABA_A_Rs on sensory afferents [93]. Our data indicate that gephyrin may be a particularly unreliable marker for PV-type inhibitory synapses and suggest that other types of inhibitory inputs be carefully evaluated for gephyrin colocalization.

### 3.2. Methodological Limitations

Critical requirements for fluorescence-based synaptic labeling tools for quantitative analysis are that (1) binding sites for labels can be fully saturated, and (2) tagging strategies do not enhance or suppress synapse stability or function. These issues have plagued first generation genetically-encoded fluorescent synapse markers that relied on overexpression of fluorescently-tagged synaptic proteins [94,95]. Although immunolabeling strategies can in theory provide uniform sample labeling, poor reagent penetration into thicker tissue specimens as well as access into the dense matrix of proteins in the postsynaptic density have reduced the efficacy of this approach. In addition, it has become increasingly clear that synaptic analysis must take into account pre- and postsynaptic identity, and antibody labeling must be integrated into multicolor fluorescence analyses. Although some studies have used labeled presynaptic terminals for quantitative input analysis [45,96], there is higher false-positive with 1- versus 2-feature (adjacent pre- and postsynaptic signal) synapse detection [3]. The addition of molecular markers localized to synapses (tagged synaptophysin or GABA coupled with Gephyrin.FingR or synaptic ultrastructure) increases the accuracy of synapse detection and improves quantitative analysis. 

We cannot exclude the possibility that Gephyrin.FingR-lacking PV synapses had low levels of Gephyrin.FingR fluorescence that we could not detect given our imaging conditions. Indeed, we found that PV boutons lacking postsynaptic gephyrin were, on average, smaller than those that were gephyrin-associated. However, we do not believe this can account for all gephyrin-negative synapses for the following two reasons. First, PV boutons lacking gephyrin were observed across the entire range of bouton sizes. Second, EM analyses of somatic synapses from fast-spiking PV neurons on Pyr neurons show that they tend to be large [37] and are likely to be resolved using light microscopy. 

PV boutons lacking postsynaptic gephyrin were observed at Pyr and PV neuron somas, and indeed the fraction of gephyrin-lacking PV terminals was higher for PV neurons. It is possible that some PV boutons at PV somas form excitatory synapses [97,98], as PV expression has been observed in glutamatergic neurons [58,59]. However, somatic inputs onto Pyr neurons are exclusively inhibitory [98,99] and PV-IPSCs were completely abolished by picrotoxin application; thus it is unlikely that gephyrin-lacking PV inputs on Pyr neurons are excitatory. Additionally, some small proportion of PV-Syn fluorescence may be non-synaptic, as previous studies have observed transport aggregates of synaptophysin in axons during early development [100,101]. However, PV-Syn fluorescence was concentrated in enlarged bouton structures adjacent to the soma of target cells, suggesting that these were indeed release sites, a conclusion supported by EM analysis. 

The Gephyrin.FingR intrabodies used here were designed to recognize trimeric gephyrin [33]. It may well be that synapses designated as gephyrin-negative in fact have low levels of gephyrin, or gephyrin in its monomeric form, although gephyrin trimerization is thought to be necessary for its synaptic localization [102,103]. Importantly, the antibodies used for immuno-EM analysis in the current study could detect monomeric gephyrin and still revealed a subpopulation of somatic GABAergic inputs that lacked immunolabeling. 

### 3.3. Pyr Neurons Show Heterogeneous Levels of Inhibition

Our study revealed a marked variability across individual Pyr neurons for somatic gephyrin puncta and PV bouton counts, densities, and summed volumes (Figure 1, Figure 5 and Figure 7). Our analysis generally avoided cells where weak expression might be attributed to low levels of AAV construct expression. Even still, we identified neurons with bright nuclear labels (indicating high expression levels of Gephyrin.FingR) but few somatic PV boutons and few gephyrin puncta, suggesting some Pyr neurons have markedly lower levels of inhibition than neighboring cells. 

These anatomical data suggest that inhibition onto cortical Pyr neurons is highly heterogeneous, consistent with previous electrophysiological studies [36,104]. Indeed, whole-cell patch clamp recordings showed a nearly 10-fold range in PV-IPSC amplitudes (Figure 8), which might be otherwise attributed to differences in recording conditions, such as the amount of dendrite retained for individual cells in the acute brain slice preparation. Based on these new anatomical data, we conclude that levels of PV inhibition are markedly different across individual Pyr neurons, even within the same layer. These differences might correspond to differences in projection target or circuit function, as has been suggested in other studies [105].

Across layers, PV-IPSCs were larger for L5 than L2/3 Pyr neurons (Figure 9), a difference accounted for by larger L5 Pyr neuron somas accommodating a larger total number of PV synapses and summed PV terminal volumes (Figure 6D and Figure 8E). Other variables like GABA_A_R subunit composition and subsynaptic distribution, and the location of dendritic PV synapses may also contribute to these functional differences and can be assessed by future investigations. 

Variability in PV inhibition was higher for L5 Pyr neurons for nearly all comparisons. These data suggest that L5 Pyr neurons are functionally heterogeneous, consistent with the molecular diversity of cell types observed there [106]. L5 Pyr neurons showed higher PV bouton counts and total bouton volume for somatic contacts, as well as PV-IPSC amplitude (Figure 6, Figure 8 and Figure 9). However, gephyrin puncta measurements did not reveal substantial differences between L2/3 and L5 Pyr neurons, suggesting that the Gephyrin.FingR signal is poorly correlated with levels of PV synaptic inhibition. 

Our prior work using a neuroligin-based postsynaptic marker [3], as well as other electrophysiological analyses, also suggest that inhibition is not uniformly distributed across Pyr neurons in the sensory cortex [36,104]. In general, fine-scale quantitative analyses have not been carried out across large populations of cells [37,98,107], and the functional implications of this diversity have not been appreciated. Future experiments will investigate whether these differences correspond to Pyr neuron identity or projection target and other properties of the cortical circuit.

### 3.4. Properties of Gephyrin-Lacking Synapses

The functional properties of gephyrin-lacking inhibitory synapses are of great interest. Gephyrin-negative synapses in the neocortex might be less mature, similar to the delayed accumulation of PSD-95 at nascent excitatory synapses [108]. Our finding that PV boutons lacking gephyrin tended to be both smaller and less intensely labeled suggests that they may represent an intermediate step in synapse formation or elimination. Indeed, it is tempting to speculate that gephyrin-lacking PV synapses may be a more labile subclass of PV synapse that can rapidly respond to changes in network activity. Prior studies indicate that PV boutons and synapses are dynamic, changing in response to learning and experience [44,45,109,110], and in vivo imaging of inhibitory synapses shows that they can repeatedly form and disappear at the same location [6]. Investigating the role of gephyrin-lacking PV synapses in inhibitory synapse development, function, and plasticity is an exciting direction for future research. 

Our results indicate that the use of gephyrin labeling as a proxy for inhibitory synapses may significantly underestimate the number of GABAergic synapses, particularly for inhibitory synapses from PV GABAergic neurons. Super-resolution imaging techniques or cryo-electron tomography will also be useful to determine how specific molecular and physiological features are related to the organization of synaptic proteins in these inhibitory synapses. In addition, the functional properties of PV terminals with low or absent gephyrin will be of great interest, particularly with regard to their GABA_A_R composition and susceptibility to experience-dependent plasticity. 

## 4. Materials and Methods

### 4.1. AAV Generation

Gephyrin.FingR construct [33] was modified for expression in mammalian neurons using the human Synapsin promoter (hSyn). The generation of the plasmid pAAV-ZFN-hSyn-Gephyrin.FingR-EGFP- IL2RGTC-KRAB-A was based on the backbone of pAAV-hSyn-LA-mNeptune2 [111]. In the first step, the zinc finger binding site nucleotides (ZFN) were inserted into the backbone to generate pAAV-ZFN-hSyn-LA-mNeptune2. Next, the sequences of the gephyrin antibody like protein (Gephyrin.FingR) and the transcriptional regulatory elements IL2RGTC-KRAB-A were PCR amplified from pCAG_Gephyrin.FingR-mKate2-IL2RGTC (gift from Don Arnold, Addgene plasmid #46297). To change mKate2 against EGFP, the sequence of EGFP was also amplified by PCR. Finally, all fragments were ligated into the backbone pAAV-ZFN-hSyn-LA-mNeptune2 digested with KpnI and HindIII to generate pAAV-ZFN-hSyn- Gephyrin.FingR-EGFP-IL2RGTC-KRAB-A. Primers and restriction endonucleases are listed in Table 1.

### 4.2. Virus Production

Recombinant adeno-associated viral (rAAV) particles with mixed serotype 1 and 2 of pAAV plasmids encoding the protein of interests were produced in HEK293-FT cells. The entire procedure was previously described in detail [111] and is applied here with the following modifications: After DNaseI treatment (30 min at 37 °C), the suspension was centrifuged at 1200 g for 10 min. The supernatant was filtered through a 0.45 µm sterile filter (Merck/Millipore, Darmstadt, Germany) and first applied to an Amicon Ultra-15, 100 kDa, centrifugal filter unit (Merck/Millipore, Darmstadt, Germany), followed by a Vivaspin^®^ 500, 100 kDa, centrifugal concentrator (Sartorius, Göttingen, Germany) to wash the virus in Opti-MEM™ Medium (ThermoFisher Scientific, Darmstadt, Germany) and concentrate to a final volume of 150 µL.

### 4.3. Animals for Gephyrin.FingR Analysis

The Gephyrin.FingR expression pattern in the barrel cortex was assessed in PV-Cre (Stock ID 008069; The Jackson Laboratory, Bar Harbor, ME, USA) [46] crossed with Cre-dependent Synaptophysin-tdTomato (Ai34; Stock ID 012570; The Jackson Laboratory, Bar Harbor, ME, USA) [42] double-transgenic knock-in animals. Animals were stereotaxically injected while under isoflurane anesthesia at weaning with rAAV1/2-ZFN-hSyn-Gephyrin.FingR-EGFP-IL2RGTC-KRAB-A virus (0.1 µL) through a small craniotomy (from bregma: x = −3, y = −0.9, z = −0.5 mm) using a Nanoject II (Drummond Scientific Company, Broomall, PA, USA).

Animals were housed in 12:12 light/dark cycle with same-sex littermates for 1–3 weeks after virus injection surgery until 48 h before tissue collection, at which time they were singly housed until tissue collection. At midday, animals were anesthetized with isoflurane and transcardially perfused using 20 mL PBS (pH 7.4), followed by 20 mL 4% paraformaldehyde in PBS (PFA, pH 7.4). Brains were removed and post-fixed in 4% PFA overnight at 4 °C before transfer to 30% sucrose (in PB, pH 7.4). After osmotic equilibration, 40 µm-thick brain sections were collected using a freezing microtome. Free-floating brain sections containing EGFP-labeled nuclei in barrel cortex were washed with PBS before mounting on microscope slides using Vectashield Antifade Mounting Medium (Vector Laboratories, Burlingame, CA, USA). A subset of sections was also immunostained for PV, vesicular acetylcholine transporter, or vesicular glutamate transporter 1, however immunofluorescence (IF) data analyses were not included in the present report. Quantitative data included in this study were collected from animals that did not undergo any additional experimental interventions and included 3 male mice, aged postnatal day (P)29–48 at 9–19-days post infection. Gephyrin.FingR expression in the female cortex was qualitatively and quantitatively similar but was not included in the current report because this animal underwent an additional experimental sensory manipulation known to induce cortical synaptic plasticity.

### 4.4. Cell Selection for Gephyrin.FingR Analysis

Pyr neurons were selected for analysis if they had EGFP nuclear labeling (indicative of nuclear transport of unbound Gephyrin.FingR) [33], and discrete punctate cell-surface EGFP labeling (indicative of synaptic Gephyrin.FingR localization). Selected Pyr neurons exhibited a prominent apical dendrite, laterally projecting basal dendrites, and a descending axon. These morphological features were apparent using cytoplasmic EGFP fluorescence and the outline of the postsynaptic cell as defined by PV-Cre axon boutons labeled with Synaptophysin-tdTomato (PV-Syn). Because of fluorescence wavelengths used for Gephyrin.FingR (EGFP) and PV axonal bouton labeling (tdTomato), we could not easily use an additional cell-filling fluorophore to facilitate morphological identification. Our analysis was biased toward Pyr neurons with a larger number of gephyrin puncta on the soma. Despite this, we found a number of neurons in both L2/3 and L5 with low numbers of somatic Gephyrin.FingR puncta. 

Construct design from Gross et al. includes a negative feedback mechanism by which excess Gephyrin.FingR is trafficked back to the nucleus to suppress further Gephyrin.FingR expression [33]. Variation in nuclear EGFP fluorescence levels suggests that nuclear localization of Gephyrin.FingR might occur before synaptic sites were fully saturated. However, Gephyrin.FingR puncta intensity was not well-correlated with nuclear labeling. There were cells with bright synaptic Gephyrin.FingR puncta at high density with dim nuclear EGFP fluorescence, as well as cells with bright nuclear EGFP fluorescence with dim synaptic Gephyrin.FingR puncta at low density. Overall, we observed no correlation between Gephyrin.FingR puncta density and intensity (Figure 2F–G). 

PV neurons were identified based on the presence of cytoplasmic Synaptophysin-tdTomato fluorescence, which was discernible above background fluorescence levels and was distinct from brighter, smaller presynaptic PV-Syn fluorescence. 

### 4.5. Gephyrin.FingR Immunolabeling Validation

Gephyrin immunolabeling was used to validate Gephyrin.FingR labeling of endogenous gephyrin in the somatosensory neocortex from C57BL/6 J mice (Stock ID 0006694; The Jackson Laboratory, Bar Harbor, ME, USA) transduced with rAAV1/2-ZFN-hSyn-Gephyrin.FingR-EGFP-IL2RGTC-KRAB-A virus. Brain sections with Gephyrin.FingR-labeled neurons were gently rocked in blocking solution for two-hours (0.25% Triton-X, 10% goat serum in PBS) before incubation with mouse monoclonal gephyrin antibody fluorescence-labeled with Oyster^®^ 550 (1:250 in blocking solution, Synaptic Systems 147 011C3, AB_887716) for 48 h at 4 °C with gentle rocking. Sections were then washed with PBS and mounted on microscope slides using Vectashield Antifade Mounting Medium (Vector Laboratories, Burlingame, CA, USA). Putative L2/3 and L5 pyramidal neurons were selected and imaged as 3D-stacks at 63× based on the following criteria: pyramidal soma with distinct apical dendrite and multiple basal dendrites, punctate staining around the soma border and along dendrites, and optimized laser and gain settings to capture the best signal-to-noise without any over- or under-saturated pixels in and around the cell of interest in either fluorescent channel. For analysis using ImageJ (version 1.53c), optical sections were chosen at or near the center of the cell body, and freehand contours were drawn to approximate the individual shapes of all somatic puncta using Gephyrin.FingR signal (12–34 puncta across cells). The same contours were then transposed to the gephyrin-IF channel. Contours of similar sizes were also drawn in background area devoid of punctate fluorescence around the cell of interest. Mean intensities of individual controls from each cell were measured and corrected for background signal. To avoid artifacts of mean intensity differences across cells, mean intensities of individual puncta from each cell were mean-scaled to the cell mean for each fluorescent channel. For correlation analysis between Gephyrin.FingR and gephyrin-IF, data from ten random puncta from each analyzed cell were taken and pooled (Figure 3).

### 4.6. Gephyrin.FingR Imaging

Confocal image stacks centered around Gephyrin.FingR-expressing soma were collected with an LSM 880 AxioObserver Microscope (Zeiss) using a 63× oil-immersion objective lens (Plan-Apochromat, 1/40 Oil DIC M27) with the zoom factor set to 1, and the pinhole set at 1.0 Airy disk unit for each fluorescence channel. Laser intensities for each channel were independently set to avoid pixel saturation for each field of view. Fluorescence acquisition setting were as follows: EGFP (excitation λ488, detection λ489–512), tdTomato and Oyster^®^ 550 (excitation λ561, detection 566–579). Maximum image size was 1024 × 124 pixels to collect 135 × 135 × ≤40 µm images, with a corresponding voxel dimension of 0.13 µm (x, y) and 0.3 µm (z). 

### 4.7. Gephyrin.FingR Image Analysis

Carl Zeiss image files were imported into Imaris (v8.4 with FilamentTracer, Bitplane. Zurich, Switzerland) for quantitative analysis. In Imaris, Gephyrin.FingR puncta were rendered as 3D objects using an experimenter-set, manual background subtraction threshold to conservatively estimate puncta borders from EGFP signals for the entire field of view (FOV) using the following settings: 0.5 µm estimated diameter, split-touching object setting with the same 0.5 µm estimated diameter, all “quality”-filter identified puncta included with a 6-voxel minimum cut-off size. EGFP-labeled nuclei were digitally subtracted from Gephyrin.FingR puncta renderings. A small fraction of puncta that were not captured using these settings were manually generated by the experimenter during examination of each optical section for concordance between fluorescence signal and 3D renderings. PV-Syn boutons were rendered using the following settings: 0.6 µm estimated diameter, split-touching object setting with a 1 µm estimated diameter, including all “quality”-filter identified puncta with a 1 µm^2^ minimum surface area size cut-off. Instead of rendering PV-Syn boutons for the entire FOV, presynaptic bouton renderings were generated for a sub-volume region of interest (ROI) just large enough to encompass the target neuron’s soma and surrounding PV-Syn boutons. Individual Gephyrin.FingR puncta and PV-Syn bouton renderings at each target neuron’s soma surface were identified using a combination of distance from the cell’s nucleus and manual selection by the experimenter. Gephyrin.FingR puncta and PV-Syn bouton alignments were digitally determined if the edges of the rendered objects were less than 0.15 µm apart. This distance was slightly below the diffraction limit of our confocal images, accounted for small gaps between pre- and postsynaptic rendered objects related to conservative border estimations, and is consistent with previous reports using fluorescent object localization [3,112]. 

### 4.8. Conjugate Immunofluorescence—SEM

The same dataset as reported in Collman at al., 2015 was used [113]. Portions are publicly available at https://neurodata.io/data/collman15/ (accessed 6 September 2021).

Briefly, a 3-month-old male mouse was perfused with 2% glutaraldehyde/2% formaldehyde in 0.1 M phosphate buffer (pH 6.8), somatosensory cortex was dissected out and small chunks of tissue were freeze-substituted and embedded in Lowicryl HM-20. Ribbons were prepared and imaged using standard methods of array tomography [48]. 70 nm-thick serial sections of the embedded plastic block were cut on an ultramicrotome (Leica Ultracut EM UC6) and mounted on carbon-coated coverslips. The sections were pretreated with sodium borohydride (1% in Tris buffered saline (TBS), pH7.6 for 3 min) to reduce non-specific staining and autofluorescence. After a 20 min wash with TBS, the sections were incubated in 50 mM glycine in TBS for 5 min, followed by blocking solution (0.05% Tween-20 and 0.1%BSA in TBS) for 5 min. The primary antibodies were diluted in blocking solution as specified in Table 2 and were applied overnight at 4 °C. After a 15 min wash in TBS, the sections were incubated with Alexa dye conjugated goat secondary antibodies, highly cross-adsorbed (Invitrogen, Waltham, MA, USA), diluted 1:150 in blocking solution for 30 min at room temperature. The sections were then washed with TBS for 15 min, rinsed with distilled water, and mounted on glass slides using SlowFade Gold Antifade Mountant with DAPI (ThermoFisher Scientific, S36937). Sections were imaged on an automated epifluorescent microscope (Zeiss AxioImager Z1) using a 63× Plan-Apochromat 1.4 NA oil objective. After imaging, the antibodies were eluted using a solution of 0.2 M NaOH and 0.02% SDS and new antibodies were reapplied. Several rounds of elution and restaining were applied to create a high-dimensional immunofluorescent image. Subsequently, the coverslips were prepared for SEM imaging by poststaining with heavy metals [113]. Sections were rinsed with water, treated for 1 min with 0.1% solution of KMnO_4_ in 0.1N H_2_SO_4_, then poststained with 5% aqueous solution of uranyl acetate for 30 min, followed by 1% Reynolds’ lead citrate for 1 min, then washed and air dried. Ribbons were imaged on a Zeiss Sigma FESEM microscope using the backscatter detector at 5–8 keV.

### 4.9. Immunofluorescent Array Tomography

All procedures related to the care and treatment of animals were approved by the Administrative Panel on Laboratory Animal Care at Stanford University. Two adult mice (3 months old), one C57BL/6 J and one YFP-H (Feng et al., 2000), were anesthetized by halothane inhalation and their brains quickly removed and placed in 4% formaldehyde in phosphate-buffered saline (PBS) at room temperature for 1 h, followed by 24 h at 4 °C in the same fixative. The somatosensory cortex was dissected out and embedded in LRWhite [113]. To preserve YFP fluorescence in the YFP-H mouse, the tissue was dehydrated only up to 70%. A series of 70-nm ultrathin sections were mounted on gelatin-coated coverslips and processed for standard indirect immunofluorescence following the same protocol as described above (4.7), except that the sections were not pretreated with sodium borohydride.

### 4.10. Image Processing and Analysis of Array Tomography

Images from different imaging sessions were registered using a DAPI stain present in the mounting medium. The images from the serial sections were also aligned using the DAPI signal. Both image registration and alignment were performed with the MultiStackReg plugin in FIJI [114]. 

In the Collman et al., 2015 dataset, inhibitory synapses were identified in scanning electron micrographs by their ultrastructural appearance (synaptic vesicles and postsynaptic thickening) and the presence of GABA immunolabel on at least 2 serial sections. Ninety-seven inhibitory synapses were identified in this dataset. An additional 102 synapses from the same sample that were imaged only with immunofluorescence were included in the final analysis. These synapses were identified by the concomitant presence of GABA, GAD and synapsin-IF on at least two serial sections. The results were not significantly different between the synapses identified by conjugate IF-SEM and those identified by immunofluorescence only. In the immunofluorescence only datasets, synapses were identified by the concomitant presence of GAD, VGAT and synapsin immunolabel on at least 2 serial sections, and immediately adjacent GABA_A_Rα1 on at least one section. They were further subdivided into PV positive and PV negative depending on the overlap of PV immunolabel with the other presynaptic markers (GAD, VGAT and synapsin).

### 4.11. Antibodies

The sources and dilutions of all primary antibodies used in this study are shown in Table 2. Secondary antibodies were Alexa Fluor 488, 594 or 647 conjugated goat secondary antibodies, IgG (H+L) highly cross-adsorbed (Invitrogen anti-mouse A11029 and A11032, anti-rabbit A11034, A11037 and A21245, and anti-guinea-pig A11073 and A21450). All secondary antibodies were diluted 1:150 in blocking solution.

### 4.12. Electrophysiology

For the functional assessment of PV-to-Pyr synaptic strength, Cre-dependent channelrhodopsin-2 (ChR2; Ai32; Stock ID 012569; The Jackson Laboratory, Bar Harbor, ME, USA) [115] and PV-Cre (Stock ID 008069; The Jackson Laboratory, Bar Harbor, ME, USA) [46] double-transgenic knock-in mice were used (male and female, P23–27). Animals were singly-housed and removed from their holding cage at midday (11 a.m.–2 p.m.) for tissue preparation.

Mice were briefly anesthetized with isoflurane before decapitation. Angled-coronal slices (45° rostro-lateral; 350 µm thick) designed to preserve columnar connections in the somatosensory cortex were prepared in ice-cold artificial cerebrospinal fluid (ACSF) composed of (in mM): 119 NaCl, 2.5 KCl, 1 NaH_2_PO_4_, 26.2 NaHCO_3_, 11 glucose, 1.3 MgSO_4_, and 2.5 CaCl_2_ equilibrated with 95%O_2_/5%CO_2_. Slices were allowed to recover at room temperature in ACSF for one hour in the dark before targeted whole-cell patch-clamp recordings were performed using an Olympus light microscope (BX51WI) and borosilicate glass electrodes (4–8 MΩ resistance) filled with internal solution composed of (in mM): 125 potassium gluconate, 10 HEPES, 2 KCl, 0.5 EGTA, 4 Mg-ATP, 0.3 Na-GTP, and trace amounts of AlexaFluor 594 (pH 7.25–7.30, 290 mOsm). Because of the need to verify cell type identity using action potential waveform, we used a K-gluconate based internal solution to enable spiking. Electrophysiological data was acquired using a MultiClamp 700B amplifier (Molecular Devices, San Jose, CA, USA), digitized with a National Instruments acquisition interface (National Instruments Corp., Austin, TX, USA), and collected using MultiClamp (Molecular Devices, San Jose, CA, USA) and IgorPro6.0 (WaveMetrics Inc., Portland, OR, USA) software with 3 kHz filtering and 10 kHz digitization. L2/3 and L5 Pyr neurons were targeted based on Pyr morphology, using the pial surface and dense PV-Ai32 fluorescence in L4 barrels for laminar orientation.

Following whole-cell break-in, presumptive Pyr cell identity was confirmed based on hyperpolarized resting membrane potential (approximately −70 mV in L2/3 and −60 mV in L5), input resistance (approximately 100–200 MΩ; <400 MΩ cut-off), and regular-spiking (RS) action potential waveforms recorded in responses to progressive depolarizing current injection steps recorded in current-clamp mode (50–400 pA, Δ50 pA steps, 0.5 s duration). L5 Pyr neurons were typically in the top to the middle portion of L5 (L5a) and had either an RS or intrinsically bursting (IB) phenotype with current injection. More than 90% of recorded cells in L5 showed an RS phenotype. Only cells with a stable baseline holding potential, resting membrane potential < −50 mV, and access resistance < 40 MΩ were analyzed [116]. Similar to previously published studies [36,41], we isolated PV-mediated IPSCs using transgenic mice expressing ChR2 in all presynaptic PV neurons using K-gluconate internal and a depolarized holding potential to evoke outward currents. We selected a holding potential (−50 mV) from the middle of their range (−45 to −55 mV). Blue light stimulation was used to evoke PV IPSCs (480 nm, 0.0065 mW per mm^2^, 5 ms pulse). Five minutes after break-in, Pyr cells were voltage-clamped at −50 mV and PV-mediated IPSCs were collected, where peak amplitude was calculated from the average of 10 sweeps (0.1 Hz). When a single light pulse evoked multiple IPSC peaks, peak IPSC amplitude was calculated for the first peak. Initially, PV IPSCs were tested at a range of LED light intensities (0.17–1.7 mW). We selected an output intensity of 0.48 mW for the following three reasons. First, evoked IPSC amplitudes were less than 1 nA to avoid introducing voltage-clamp errors. Second, 100% of trials elicited an IPSC in the recorded Pyr neuron. Third, direct recording from PV neurons indicated that this power was sufficient to drive at least one spike in >75% of cells. Consistent with a chloride-mediated current, the reversal potential for optically evoked currents was experimentally determined to be −78 ± 4 mV (uncorrected for junction potential). For a subset of cells, picrotoxin (50 µM) was bath-applied to confirm optically evoked IPSCs were mediated by GABA_A_Rs, and in all cases picrotoxin completely abolished light-evoked currents.

### 4.13. Statistical Analysis

All values reported in text and represented in graphs are mean ± SD, unless otherwise noted. Soma density is total puncta or bouton count divided by soma surface area. Laminar differences in Gephyrin.FingR and PV-Syn soma associations, and PV-IPSCs were assessed for statistical significance using the Mann–Whitney *U* test, Pearson’s correlation was used for all correlation analyses (Pearson’s R reported in figures). The Kolmogorov–Smirnov Test was used to compare cumulative distribution of Gephyrin.FingR puncta and PV bouton volumes between alignment categories across cells, and Wilcoxon signed-ranks test was used to compare Gephyrin.FignR puncta and PV bouton volumes between alignment categories within cells (OriginPro 2018, Northampton, MA, USA). Statistical significance, *p* < 0.05.

## Figures and Tables

**Figure 1 ijms-22-10032-f001:**
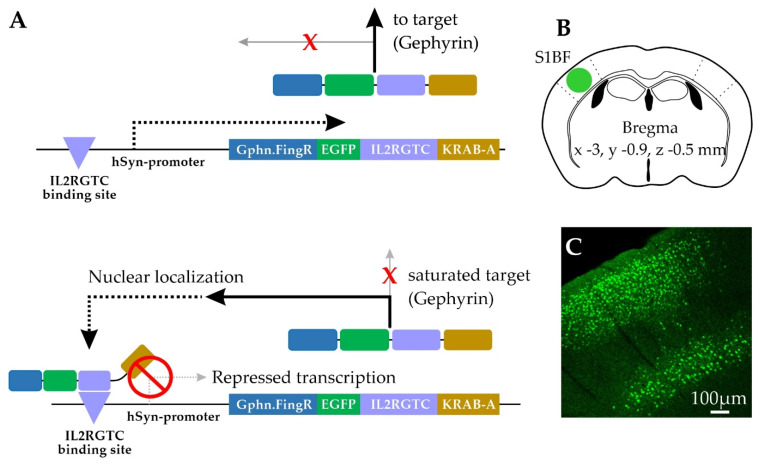
Zinc finger mediated transcriptional regulation of Gephyrin.FingR. (**A**) Adapted from Gross et al., 2013 [33]. The Gephyrin.FingR (Gphn.FingR) itself is fused to EGFP, the zinc finger (IL2RGTC), and the transcriptional repressor (KRAB-A). The zinc finger binding site was inserted upstream of the human synapsin promoter (hSyn). (**Top**) In the unsaturated state, Gephyrin.FingR is transcribed in the nucleus (dotted arrow), then the translated Gephyrin.FingR fusion protein binds to endogenous gephyrin in the cytoplasm (solid arrow). (**Bottom**) When all endogenous gephyrin is labeled with Gephyrin.FingR, the excess Gephyrin.FingR migrates back to nucleus due to a nuclear localization sequence that is part of the zinc finger domain. Back in the nucleus (dotted arrow), the fusion protein binds to the promoter via the zinc finger IL2RGTC and the transcriptional repressor KRAB-A prevents further transcription. Thus, the expression of Gephyrin.FingR is matched to the level of endogenous unsaturated gephyrin; (**B**) virus injection site coordinates for targeting primary somatosensory cortex barrel fields (S1BF); (**C**) low magnification image of transduced S1BF injection site showing L2/3 and L5 neurons with EGFP labeled nuclei.

**Figure 2 ijms-22-10032-f002:**
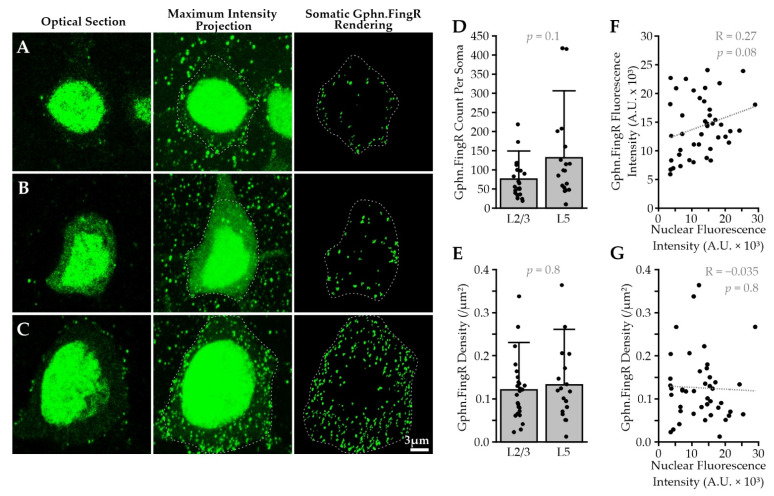
Neocortical Pyr neurons show heterogeneous levels of Gephyrin.FingR puncta. (**A**) From left-to-right: optical section of L2/3 Pyr neuron transduced with Gephyrin.FingR encoding AAV, maximum intensity projection of the same cell with soma outlined with a dotted grey line, 3D rendering of the cell’s somatic Gephyrin.FingR puncta (Gphn.FingR) with same dotted grey line outline overlaid; (**B**,**C**) as in (**A**), but for a different L2/3 and L5 Pyr neuron, respectively; (**D**) Gephyrin.FingR puncta counts per Pyr soma; (**E**) somatic Gephyrin.FingR density; (**F**) relationship between Gephyrin.FingR puncta and nuclear fluorescence; (**G**) relationship between Gephyrin.FingR density and nuclear fluorescence. Bars represent mean, error bars SDs, individual data points overlaid. L2/3: *n* = 23–25 cells (data from 2 cells with incomplete soma imaging were excluded from (**D**)); L5: *n* = 18 cells from; *N* = 3 animals.

**Figure 3 ijms-22-10032-f003:**
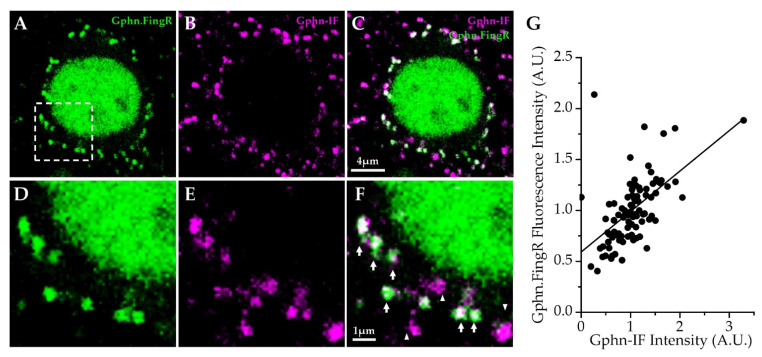
Correlated Gephyrin.FingR (Gphn.FingR) and gephyrin immunofluorescence (Gphn-IF) in neocortical Pyr neurons from mouse brain. (**A**) Optical section showing virally-transduced Gephyrin.FingR (green) puncta on a L2/3 Pyr neuron soma; (**B**) gephyrin-IF (magenta) signal for the same optical section in (**A**); (**C**) Gephyrin.FingR and gephyrin-IF for the same optical section in (**A**); (**D**–**F**) magnified images showing somatic signal corresponding to the boxed area in (**A**); arrows mark co-labeled puncta; arrowheads mark gephyrin-IF puncta that are not colocalized with Gephyrin.FingR. Since only a small fraction of cortical neurons express high levels of Gephyrin.FingR, the majority of inhibitory synapses in the brain show gephyrin-IF only; (**G**) Correlation plot for cell-normalized mean intensities of somatic Gephyrin.FingR and gephyrin-IF signal in L2/3 (*n* = 4 cells) and L5 Pyr neurons (*n* = 5 cells) from one animal.

**Figure 4 ijms-22-10032-f004:**
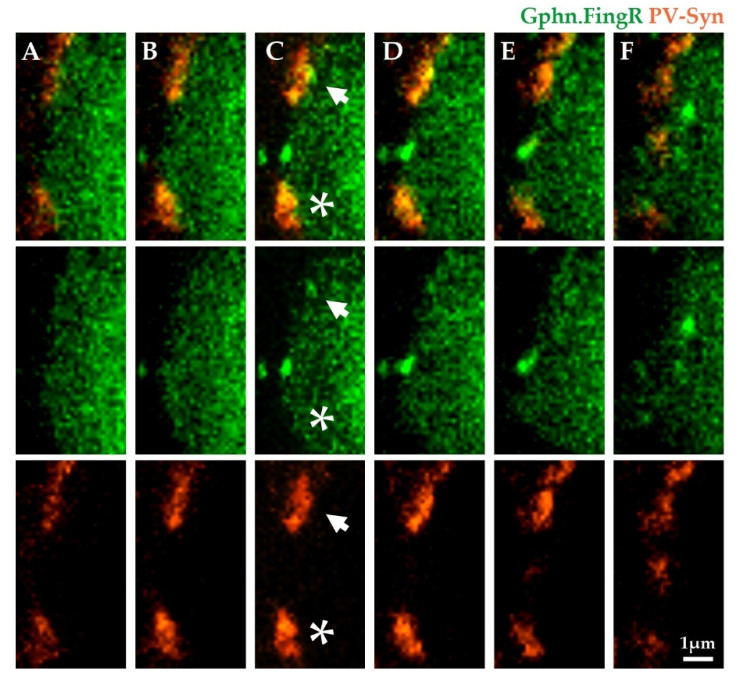
A subset of PV boutons on Pyr neuron somas lack gephyrin. (**A**–**F**) Serial optical sections from a single Pyr neuron’s soma with Gephyrin.FingR (green) and PV-Syn (orange); arrows mark bouton associated with Gephyrin.FingR, asterisks mark bouton lacking postsynaptic Gephyrin.FingR. Nucleus is at right in the image.

**Figure 5 ijms-22-10032-f005:**
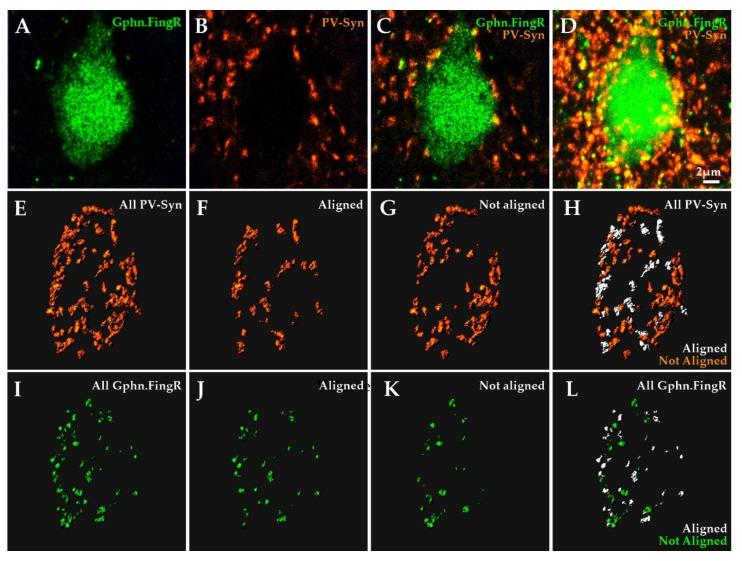
Object-based image analysis approach for quantitative assessment of Gephyrin.FingR and PV bouton alignment; (**A**–**C**) optical section of a Gephyrin.FingR (green) labeled L2/3 Pyr neuron soma and surrounding PV-Syn bouton fluorescence (orange); (**D**) maximum intensity projection of the same cell; (**E**) 3D rendering of all soma-adjacent PV-Syn boutons; (**F**) subset of boutons in **E** that aligned with Gephyrin.FingR (≤0.15 µm apart, edge-to-edge); (**G**) subset of boutons in E that did not align with Gephyrin.FingR (>0.15 µm apart); (**H**) all PV boutons color-coded based on their alignment with postsynaptic Gephyrin.FingR; (**I**) 3D rendering of all somatic Gephyrin.FingR; (**J**) subset of Gephyrin.FingR in I that could be aligned with PV boutons; (**K**) subset of Gephyrin.FingR puncta in **I** that did not align with PV boutons; (**L**) all Gephyrin.FingR puncta color-coded based on their alignment with presynaptic PV boutons.

**Figure 6 ijms-22-10032-f006:**
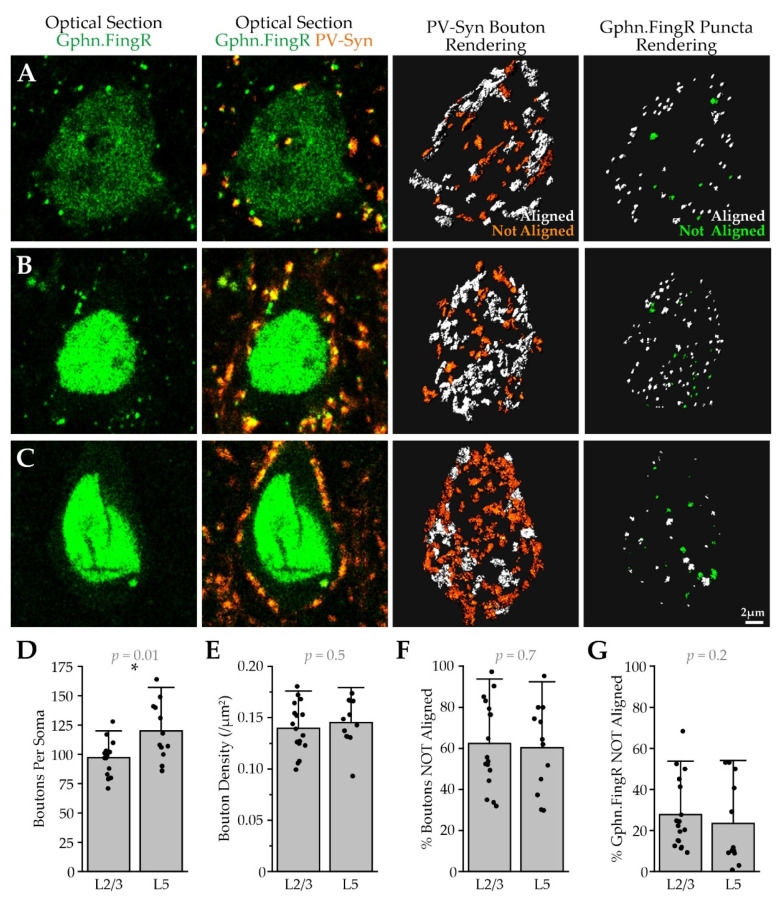
Gephyrin-lacking PV boutons are common on Pyr neuron somas. (**A**) left-to right: optical section of a Pyr neuron soma showing Gephyrin.FingR (green), PV-Syn (orange), dual-channel fluorescence, PV bouton renderings for the cell’s entire soma color-coded by gephyrin alignment, gephyrin renderings color-coded by PV bouton alignment; (**B**,**C**) as in (**A**), but for two different Pyr neurons; (**D**) the number of PV boutons touching a Pyr neuron soma, * *p* < 0.05; (**E**) PV boutons at Pyr neuron somas; (**F**) the percentage of somatic PV boutons that could not be aligned with a postsynaptic gephyrin puncta; (**G**) the percentage of somatic gephyrin puncta that could not be aligned with presynaptic PV boutons; L2/3, *n* =17 cells; L5, *n* = 12 cells; *N* = 2 animals.

**Figure 7 ijms-22-10032-f007:**
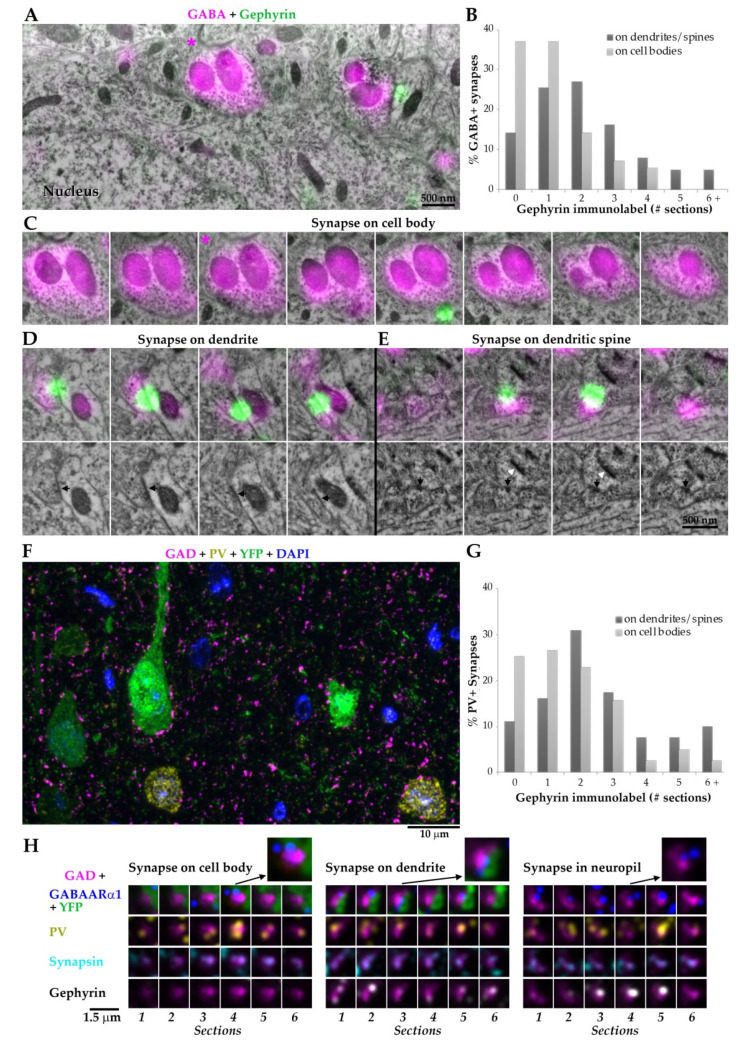
GABAergic synapses on Pyr somas have low gephyrin content. (**A**) Conjugate immunofluorescence—scanning electron micrographs (SEM) from adult mouse neocortex immunolabeled with GABA (magenta) and gephyrin (green). On this single section (70 nm), the GABA positive synapse onto the cell body (marked with asterisk) does not have postsynaptic gephyrin immunolabel, but the GABA+ synapse to the right that contacts a dendrite is gephyrin immunopositive; (**B**) gephyrin content of GABA positive synapses, depending on their target. Serial sections through the synapses were analyzed (*n* = 57 on cell bodies, and *n* = 142 on dendrites and spines); (**C**) serial sections through the somatic synapse in A, marked with asterisk. Gephyrin immunolabel was present on only one section; (**D**) serial sections through a synapse targeting a dendrite. The second row shows the SEM images only, to show the synaptic contact (black arrow) that is obscured by the bright gephyrin-IF; (**E**) serial sections through a synapse on a dendritic spine. The same spine also receives a non-GABA excitatory synapse (white arrowhead). (**F**) Immunofluorescence of neocortical layer 5 of an adult YFP-H mouse, one 70 nm section. Pyramidal neurons are identified by the native YFP fluorescence (without antibody labeling, green) and PV interneurons by immunolabeling with antibodies against PV (yellow). GAD2 immunolabel (magenta) concentrates in inhibitory synapses and nuclei are labeled with DAPI (blue) (**G**) Gephyrin content of PV synapses, depending on their target (*n* = 83 on cell bodies, and *n* = 81 on dendrites and spines, from 2 animals). (**H**) Examples of PV-positive synapses onto different targets. Columns are serial sections, and rows are different immunostains. Arrows point to higher magnification views of individual sections that contained the apposition of inhibitory presynaptic markers (GAD2, magenta; PV, yellow; synapsin, cyan; gephyrin, white) with the postsynaptic GABA_A_Rα1 (blue) required to identify synapses.

**Figure 8 ijms-22-10032-f008:**
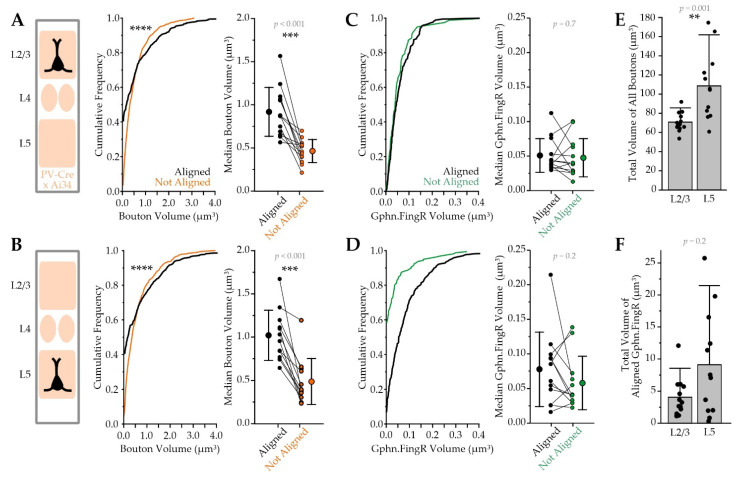
Gephyrin-lacking PV synapses have smaller boutons. (**A**) left: cortical column schematic; middle: cumulative frequency plot comparing L2/3 Pyr neuron somatic PV-Syn bouton volume for boutons that were either *aligned* with postsynaptic Gephyrin.FingR (black line) or *not aligned* (orange line; Kolmogorov-Smirnov Test, *p* < 0.0001); right: L2/3 Pyr cells’ median PV bouton volume for Gephyrin.FingR *aligned* and *not aligned* boutons with values from the same cell connected by lines and group averages (mean ± SD) to the left and right of individual cell values, paired comparison performed using Wilcoxon Signed Ranks Test; (**B**) as in A, but for L5 Pyr neurons; (**C**) left: cumulative frequency plot of L2/3 Pyr neuron somatic Gephyrin.FingR volumes for puncta that were *aligned* with PV boutons (green line) or *not aligned* (black line; Kolmogorov-Smirnov Test, *p* = 0.4), right: L2/3 Pyr cells’ median Gephyrin.FingR puncta volume for *aligned* and *not aligned* puncta with values from the same cell connected by lines and group averages (mean ± SD) to the left and right of individual cell values, paired comparison performed using Wilcoxon Signed Ranks Test; (**D**) as in (**C**), but for L5 Pyr neurons (Kolmogorov-Smirnov Test, *p* = 0.2); (**E**) total volume of all PV boutons associated with individual L2/3 and L5 Pyr neuron somas; (**F**) total volume of all Gephyrin.FingR puncta that could be aligned with PV boutons for individual L2/3 and L5 Pyr neuron somas; ** *p* ≤ 0.01, *** *p* ≤ 0.001, **** *p* ≤ 0.0001; L2/3, *n* = 13 cells; L5, *n* = 12 cells; *N* = 2 animals.

**Figure 9 ijms-22-10032-f009:**
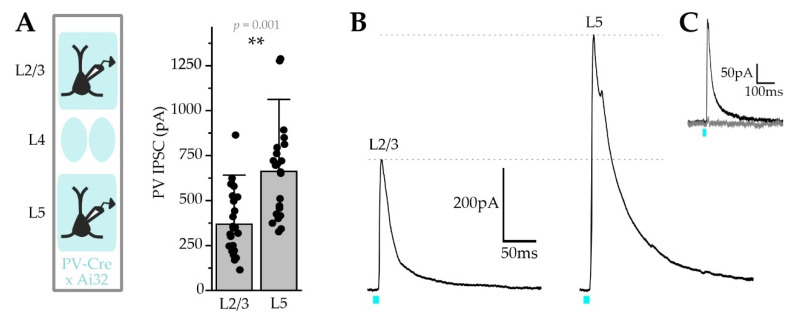
PV inhibition of L5 Pyr neurons is stronger than L2/3 Pyr neuron inhibition. (**A**) Left, schematic of recording. Right, PV-IPSC peak amplitude for L2/3 and L5 Pyr cells, ** *p* ≤ 0.01; (**B**) Example trace of a PV-IPSC from a single L2/3 (**left**) and L5 (**right**) Pyr neuron following blue-light stimulation (5 ms, blue tick mark); (**C**) A PV-IPSC recorded before (black) and after bath application of picrotoxin (grey); L2/3, *n* =24 cells; L5, *n* = 22 cells; *N* = 4 animals.

**Figure 10 ijms-22-10032-f010:**
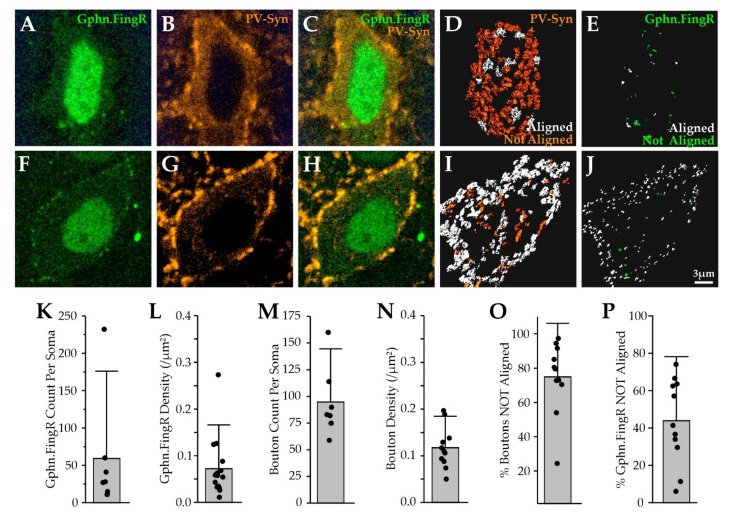
The majority of PV inputs onto PV neurons lack postsynaptic gephyrin. (**A**) optical section of a PV neuron soma showing Gephyrin.FingR (green) fluorescence; (**B**) PV-Syn fluorescence (orange) for the same cell in (**A**) showing cytoplasmic PV-Syn fluorescence; (**C**) Gephyrin.FingR and PV-Syn fluorescence overlaid; (**D**) PV bouton renderings for the cell’s entire soma, color-coded by Gephyrin.FingR alignment; (**E**) Gephyrin.FingR puncta renderings for the cell’s entire soma, color-coded by PV bouton alignments; (**F**–**J**) as in **A**–**E**, but for a different PV neuron; (**K**–**P**) quantitation of Gephyrin.FingR and PV boutons on PV somas; (**O**) percentage of PV boutons *not aligned* with postsynaptic Gephyrin.FingR; (**P**) percentage of Gephyrin.FingR puncta *not aligned* with presynaptic PV boutons.

**Table 1 ijms-22-10032-t001:** The plasmid pAAV-ZFN-hSyn-Gephyrin.FingR-EGFP- IL2RGTC-KRAB-A was cloned in two steps.

Steps	Plasmid	Primer	DNA Insert	Endonuclease
Start	pAAV-hSyn-LA-mNeptune2 ^1^			
1.	pAAV-ZFN-LA-mNeptune2	P-5′-CGCGTCTTCCACAGAGTGTGT-3′	ZFN ^2^	MluI
		P-5′-CTAGACACACTCTGTGGAAGA-3′		XbaI
2.	pAAV-ZFN-hSyn-Gephyrin.FingR-EGFP-IL2RGTC-KRAB-A	5′-TCATAGATCTGTGAGCAAGGGCGAGGAGC-3′	EGFP	BglII
		5′-GACTAGATCTCTTGTACAGCTCGTCCATGC-3′		
		5′-GAATTCGGTACCGCGGGCCCGGGA-3′	GPHN-FingR-mKate2-IL2RGTC-KRAB-A ^3^	KpnI BglII HindIII
		5′-GCGCCAAGCTTGCTTTACTTGTACGCTAA-3′		

P-5′: 5-prime phosphorylated. ^1^ Wegner et al., 2017 [111]. ^2^ Template: pCAG_Gephyrin.FingR-mKate2-IL2RGTC0. ^3^ ZFN: Zinc finger binding site (nucleotides).

**Table 2 ijms-22-10032-t002:** Primary Antibodies.

Antigen	Host	Antibody Source	Dilution	RRID
Gephyrin	mouse	BD Biosciences 612632	1:100	AB_399669
Gephyrin	mouse	NeuroMab 75–443	1:100	AB_2636851
Gephyrin	mouse	Synaptic Systems 147 011C3	1:250	AB_887716
GABA	guinea pig	Millipore AB175	1:5000	AB_91011
GABA_A_Rα1	Mouse	NeuroMab 75–136	1:100	AB_2108811
GAD2	rabbit	Cell Signaling 5843	1:200	AB_10835855
Parvalbumin	rabbit	SWANT PV27	1:300	AB_2631173
Synapsin	rabbit	Cell Signaling 5297	1:200	AB_ 2616578
VGAT	mouse	Synaptic Systems 131 011	1:100	AB_887868

## Data Availability

Conjugate IF SEM data is publicly available at https://neurodata.io/data/collman15/ (accessed on 6 September 2021). Gephyrin.FingR expressing tissue data is available upon request.

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
