# Peer review of "Gephyrin-Lacking PV Synapses on Neocortical Pyramidal Neurons"

_ijms, 2021, doi:10.3390/ijms221810032_

Round 1

Reviewer 1 Report

The manuscript provides molecular and histological evidence for the existence inhibitory synapses devoid of the scaffolding protein gephyrin in the mouse somatosensory cortex. Indeed, this protein is classically used to label inhibitory synapse in many brain regions but many data have questioned the absolute requirement of gephyrin at GABergic synapses. In this study the authors used a nice approach by of specific intrabody labeling endogenous gephyrin to examine its alignment with axon terminal deriving from PV expressing GABAergic interneurons. They used AAV vector to express GFP coupled intrabody and localized gephyrin. Overall the data are quite convincing but I have several point I would like the authors to bring more information or data.

  1. In the abstract, line 24, the study focuses only on PV to pyr connection. This sentence should be changed.
  2. Page 3. Please add a drawing of AAV1/2-ZFN-hSyn-Gephyrin.FingR-EGFP-IL2RGTC-KRAB-A , the general strategy, and explain more how it works in the cell with the transcriptional regulation (see methods). Reference should be indicated at this point. Several vectors are mentioned (line 101) but only one is described. Please clarify. An image of the injection site would be welcome.
  3. Page 3. In all figure legends, panel are labeled with uppercase in the figure and lowercase in the legend.
  4. Page 3. Please explain with the nuclear fluorescence is an indicator of saturating level of Gephyrin.FingR.
  5. Page 4. Figure 2. It is unclear whether panel D and H refer to a single cell or several cells. How the fluorescence intensity has been quantified since light has been acquired is two separated channel and setting are adjusted independently apparently. It is written in methods that a SStCre mouse line has been used in combination with Cre dependent AAV expressing Gephyrin.FingR-EGFP. Please clarify. The validation can’t be performed on SSt GABAergic terminal while the rest of the study focuses on PV to Pyr synapses.
  6. Page 5 line 168. Please explain what you mean by digital rendering. The workflow is not properly described in figure 4.
  7. Page 6 line 191 please explicit what you mean by “numbers”. In the section to do not give results as mean +/- SD by just approximate.
  8. Page 8 line 237, please define “SEM image”.
  9. Page 9 in the panel A, authors refers to AI34, please clarify in the results section. Based on method it is the line used in the whole study for labeling PV terminal but there is no reference to the AI34 line the text except in the methods.
  10. Page 10. Authors used optogenetics to record IPSC from PV interneurons. They found that current record in L5 where larger than those in L2/3 pyr neurons. However, these experiments where led without clear calibration. This difference could be due to many factors other than the size of PV terminals such as density of terminal, GABA receptor composition and of course light intensity.
  11. Page 15. Authors should describe more in detail how they generated the plasmid for AAV production. The table 1 is not refered in the text. They have generated a Cre dependent AAV but I don’t see the link with the rest of the study.
  12. Page 16, please describe the protocol for immunolabeling. The anibody used for labeling GFP is not given in table 2.
  13. Page 17 line 609. Please explain the choice of this threshold (0.15µm)
  14. Page 18. In table 2 I don’t see any reference in the results section of the study about immunolabeling against GAD2 or synapsin. Authors should provide information regarding secondary antibodies in this table.

Author Response

Reviewer 1

We appreciate comments from Reviewer 1, who indicated that the data were “quite convincing” but requested some additional data and experiments to strengthen the paper.  We have now added two new figures to address specific concerns regarding construct function and gephyrin.FingR validation with immunohistochemistry in target Pyr neurons, and have updated all reagent tables to specify antibodies used.  We hope that these revisions have substantially improved the manuscript.  Responses to specific concerns are listed below.

  1. In the abstract, line 24, the study focuses only on PV to pyr connection. This sentence should be changed.

Since we investigated both PV onto Pyr as well as PV onto PV synapses, the wording has been slightly modified (line 23-34):

“Here we investigated input- and target-specific localization of gephyrin at a defined class of inhibitory synapses, using Gephyrin.FingR proteins tagged with EGFP in brain tissue from transgenic mice.” 

2. Page 3. Please add a drawing of AAV1/2-ZFN-hSyn-Gephyrin.FingR-EGFP-IL2RGTC-KRAB-A , the general strategy, and explain more how it works in the cell with the transcriptional regulation (see methods). Reference should be indicated at this point. Several vectors are mentioned (line 101) but only one is described. Please clarify. An image of the injection site would be welcome.

A diagram of transcriptional regulation, drawing of AAV1/2-ZFN-hSyn-Gephyrin.FingR-EGFP-IL2RGTC-KRAB-A, as well as a diagram and image of the injection site have been added to new Figure 1. In addition, we have modified the text to include a more detailed description of how transcriptional regulation works and reference to Gross et al., 2013:

3. Page 3. In all figure legends, panel are labeled with uppercase in the figure and lowercase in the legend.

Revised as suggested.

4. Page 3. Please explain with the nuclear fluorescence is an indicator of saturating level of Gephyrin.FingR.

We have revised the text as follows (line 98-108):

“Gephyrin.FingR expression is controlled by negative feedback regulation so that once endogenous gephyrin binding sites are occupied, unbound Gephyrin.FingR is trafficked to the nucleus due to a nuclear localization sequence that is part of the IL2RGTC zinc finger domain (Figure 1A).  Binding of the repressor KRAB-A to the promoter via the zinc finger IL2RGTC inhibits further transcription and expression of Gephyrin.FingR [33]. By this means excess background fluorescence in the dendrites is reduced and nuclear fluorescence is an indicator of saturating levels of Gephyrin.FingR.”

5. Page 4. Figure 2. It is unclear whether panel D and H refer to a single cell or several cells. How the fluorescence intensity has been quantified since light has been acquired is two separated channel and setting are adjusted independently apparently. It is written in methods that a SStCre mouse line has been used in combination with Cre dependent AAV expressing Gephyrin.FingR-EGFP. Please clarify. The validation can’t be performed on SSt GABAergic terminal while the rest of the study focuses on PV to Pyr synapses.

Figure 2 had showed validation of gephyrin immunofluorescence colocalization with Gephyrin.FingR in SST neurons, not at SST terminals. The presynaptic terminals adjacent to these gephyrin puncta in SST cells were not labeled. However, we agree showing validation in SST neurons is not ideal since other figures focus on Pyr and PV neurons. We have now carried out additional experiments and analysis in Pyr neurons which is now shown in a replacement figure, Figure 3.  These data represent gephyrin puncta from the soma of multiple L2/3 and L5 Pyr neurons. Importantly, image collection parameters were set to avoid under- or over-saturating pixels in regions of interest for both fluorescence channels. As the reviewer notes, the absolute fluorescence intensities of each channel cannot be meaningfully compared because they required independent adjustment, cell-to-cell. Nevertheless, relative fluorescence intensity within cells should show a consistent pattern across labeling methods/channels if they similarly report endogenous gephyrin protein abundance. To perform relative fluorescence intensity calculations, we performed background correction and mean-scaled the intensity data for all puncta from individual cells before pooling across cells for correlation analysis. Analysis in new Figure 3 shows that gephyrin.FingR and gephyrin-immunofluorescence are highly correlated.

Note that we have tried immunostaining using multiple other gephyrin and GABAAR subunit targeting antibodies (NeuroMab Gephyrin antibody 73-443; Synaptic Systems Gephyrin 147 011; Synaptic Systems Gephyrin Oyster650; Millipore GABAAR α1 06868; Millipore GABAAR γ2, AB5559), but only observed reliable immunofluorescence signal using Synaptic System’s gephyrin-oyster550 conjugated antibody. Since presynaptic PV-Cre x Synaptophysin-tdTomato and postsynaptic gephyrin-oyster550 conjugated antibody’s fluorescent proteins’ excitation and emission wavelengths overlap substantially, we were unable to perform this validation experiment in our Gephyrin.FingR-expressing tissue.

6. Page 5 line 168. Please explain what you mean by digital rendering. The workflow is not properly described in figure 4.

We appreciate the value of a more complete description of the analysis workflow description and have now expanded this section, including additional explanation for digital rendering and analysis terminology beginning at line 194:

“The frequency of Gephyrin.FingR-lacking PV boutons was quantitated using an object-based image analysis approach (Figure 4). Based on resolution limits of conventional confocal microscopy, we expected some overlap or adjacency between pre- and postsynaptic structures for any given synapse, and thus used a distance threshold (<0.15 µm) between edges of 3D digital renderings of PV boutons and gephyrin puncta to assess alignment. Boutons adjacent to or touching gephyrin puncta were classified as aligned with gephyrin (Figure 4F), and those beyond the threshold were classified as not aligned (Figure 4G, H). The same approach was used to classify all gephyrin puncta (Figure 4I) as aligned with PV boutons (PV synapses; Figure 4J), or not aligned (non-PV synapses; Figure 4K,L).“

7. Page 6 line 191 please explicit what you mean by “numbers”. In the section to do not give results as mean +/- SD by just approximate.

The following sentence has been modified to disambiguate meaning of numbers (line 223-225):  

“A positive correlation between gephyrin puncta and PV bouton numbers per cell might suggest these synaptic markers might be related if not spatially coupled.”

8. Page 8 line 237, please define “SEM image”.

The figure caption has been updated to include the definition for the SEM acronym.

9.  Page 9 in the panel A, authors refers to AI34, please clarify in the results section. Based on method it is the line used in the whole study for labeling PV terminal but there is no reference to the AI34 line the text except in the methods.

Reference to Ai34 has been added to line 169:

“To label presynaptic PV boutons, we employed Cre-dependent expression of tdTomato-tagged synaptophysin (Syn) using the Ai34 line crossed with PV-Cre animals (PV-Syn) and examined their distribution relative to postsynaptic Gephyrin.FingR puncta using volumetric confocal imaging.”

10. Page 10. Authors used optogenetics to record IPSC from PV interneurons. They found that current record in L5 where larger than those in L2/3 pyr neurons. However, these experiments where led without clear calibration. This difference could be due to many factors other than the size of PV terminals such as density of terminal, GABA receptor composition and of course light intensity.

We agree that proper calibration of the light intensity is critical for interpretation of these experiments.  This process is described in the methods section (lines 811-816). Importantly, the same intensity of light was used for L2/3 and L5 recordings enabling comparison of evoked current magnitude across lamina.

We appreciate that multiple factors may contribute to observed differences in PV IPSCs in L2/3 versus L5. We have added these suggested possibilities to the discussion as avenues for future investigation in lines 523-528.

11. Page 15. Authors should describe more in detail how they generated the plasmid for AAV production. The table 1 is not refered in the text. They have generated a Cre dependent AAV but I don’t see the link with the rest of the study.

We apologize for the lack of clarity in description of AAV construction.  These details have now been added to the Methods and Table 1 Additional detail for plasmid AAV production and reference to Table 1 has been added to lines 565-577:

12. Page 16, please describe the protocol for immunolabeling. The anibody used for labeling GFP is not given in table 2.

Table 2 is now updated to include additional primary antibodies used in this study. The secondary antibodies are specified in the text (4.10. Antibodies).

The immunolabeling protocol for conjugate immunofluorescence – SEM has been also been added to lines 706-734.

13. Page 17 line 609. Please explain the choice of this threshold (0.15µm)

We have now included a more complete explanation for selecting this threshold, added to lines 703-705:

“In short, this distance accounted for the small gaps between pre- and postsynaptic-rendered objects related to conservative border estimations, was slightly below the diffraction limit of our confocal images, and was consistent with previous studies using fluorescent object localization [3,110].”

14. Page 18. In table 2 I don’t see any reference in the results section of the study about immunolabeling against GAD2 or synapsin. Authors should provide information regarding secondary antibodies in this table.

We apologize for the omission.  GAD2 and synapsin antibodies are now referenced in the Results section and in Figure 6H. Several more antibodies are now also included in Table 2 as they are referenced in Methods 4.9. Image processing and analysis of array tomography (VGAT, PV and GABAARa1) and Figure 6 (PV and GABAARa1).

Reviewer 2 Report

The Introduction of this paper is very well laid out and the experiments were in general carefully carried out and analyzed.  However, the main conclusion stated in the Abstract that “a substantial population of inhibitory synapses operate without gephyrin” (line 30) may need more definitive proof, and a statement in line 88 that “gephyrin is not a required constituent of PV inhibitory synapse” could be toned down.

Specifically,

  1. how specific is this Cre-dependent tdTomato-tagged synaptophysin signal as a marker for presynaptic terminal? It has been shown that in young neurons, synaptophysin is robustly transported in axons as aggregates without being localized at synaptic sites.  What if these synaptophysin puncta of PV axons are “transport aggregates” instead of synaptic vesicle clusters at presynaptic sites? This possibility should be discussed, especially that the age of animals was at a relatively young age (p 29-48), where transport aggregates could be a possibility.
  2. 2 is meant to show “a strong correlation” between Gephyrin.FingR and gephyrin IF, but there clearly are many more red puncta (gephyrin IF) than those colocalized with green puncta (Gphn.FingR). Although the last sentence in figure legend attributed these red punta as inhibitory synapses of untransduced neurons, there are still a few red dots in (G) that appear to be associated with the green soma but not co-localized with green puncta.  Is it possible that some inhibitory postsynaptic specializations are composed of endogenous gephyrin only?
  3. In the same vein, is it possible to do Gephyrin IF on materials shown in Fig. 4 &5? In these two main figures of this paper, alignment of Gephn.FingR and PV-Syn was assessed, and the authors showed a substantial PV-Syn boutons without apposing Gephn.FingR. If immunolabeling of Gephyrin antibody can also substantiate this finding, it would provide even stronger support.
  4. Has it been done in WT cell culture systems of triple labeling of (1) PV interneuron axons, (2) presynaptic markers (either synaptic vesicle or active zone markers like bassoon or piccolo) and (3) gephyrin?  Could this provide another way to prove PV boutons without gephyrin signal?

Another MAJOR concern is on section 2.4 “correlated light and EM….”

Fig. 6 is based on ONE WT animal perfusion-fixed with 2% glutaraldehyde plus 2% PF, which is a rather heavy fixation that may reduce the antigenicity of target proteins to be immunolabeled. If the labeling efficiency is compromised by this fixation, then only a fraction of the antigen would be labeled.  Lack of gephyrin labeling at some inhibitory synapses could simply be due to low labeling sensitivity.  The subtitle of Fig. 6 should be changed to “GABAergic synapses have lower gephyrin content on Pyr somas than on dendrites”.  In fact, this reviewer suggests that the entire section 2.4 be deleted because this line of evidence is weak, and the material used is totally different from the rest of the paper.

__

Additional MINOR point,

  1. Figure panels are labeled with upper case, figure legends are in lower case.
  2. It was obvious in Fig. 3 that there is gephyrin punta unopposed by PV-Syn. The explanation appeared much later in line 194,195 that they may be linked to other non-PV interneurons. Perhaps this explanation could appear earlier in association with Fig. 3.
  3. 6E may not be a spine, could be a thin dendrite shaft. Has it ever been shown that a spine can receive both excitatory and inhibitory inputs?  The cytoplasmic content of this “spine” appears more like dendritic shaft.
  4. There are two “data not shown” (line 254, line 289) that could be either supplemented with additional info, e. g., (n) values in the text, or provided in supplementary data.

Author Response

Reviewer 2:

We greatly appreciate comments and suggestions from Reviewer 2 and in response to concerns raised, we have added two new figures as well as two new supplemental figures to the revised manuscript.  We have also extensively revised the EM figure to include new data from a complementary embedding and staining method, to alleviate concerns about antigen detection.  Below please find specific responses to concerns raised in the initial review.

Overview:

The Introduction of this paper is very well laid out and the experiments were in general carefully carried out and analyzed.  However, the main conclusion stated in the Abstract that “a substantial population of inhibitory synapses operate without gephyrin” (line 30) may need more definitive proof, and a statement in line 88 that “gephyrin is not a required constituent of PV inhibitory synapse” could be toned down.

We have modified the wording in the abstract to more accurately represent our findings as follows:

 (line 29-30) “Our findings suggest some inhibitory synapses may lack gephyrin.” 

(line  88-90) “These data indicate that gephyrin may not be a required constituent of PV inhibitory synapses and suggest that gephyrin-lacking PV synapses may play a distinct functional role.”

  1. how specific is this Cre-dependent tdTomato-tagged synaptophysin signal as a marker for presynaptic terminal? It has been shown that in young neurons, synaptophysin is robustly transported in axons as aggregates without being localized at synaptic sites.  What if these synaptophysin puncta of PV axons are “transport aggregates” instead of synaptic vesicle clusters at presynaptic sites? This possibility should be discussed, especially that the age of animals was at a relatively young age (p 29-48), where transport aggregates could be a possibility.

We appreciate that synaptophysin may not be exclusively found at release sites, and a possible contribution of non-synaptic axonal synaptophysin aggregates to synapse identification error rate has been added to the discussion (line 496-498):

“Additionally, some small proportion of PV-Syn fluorescence may be non-synaptic, as previous studies have observed transport aggregates of synaptophysin in axons during early development [98,99].  However, PV-Syn fluorescence was concentrated in enlarged bouton structures adjacent to the soma of target cells, suggesting that these were indeed release sites, a conclusion supported by EM analysis.”

2. Figure 2 is meant to show “a strong correlation” between Gephyrin.FingR and gephyrin IF, but there clearly are many more red puncta (gephyrin IF) than those colocalized with green puncta (Gphn.FingR). Although the last sentence in figure legend attributed these red punta as inhibitory synapses of untransduced neurons, there are still a few red dots in (G) that appear to be associated with the green soma but not co-localized with green puncta.  Is it possible that some inhibitory postsynaptic specializations are composed of endogenous gephyrin only?

Based on Gross et al.’s more extensive characterization of the gephyrin intrabody in low density dissociated cortical cultures with good isolation of postsynaptic cells, very strong concordance between intrabody labeling and endogenous gephyrin distribution was observed. It remains possible that due to an unknown mechanism, intrabody accessibility to specific synapses subtypes within a cell may be limited. We note that based upon concerns from reviewer 1 about analysis in SST neurons, to ensure continuity of cell type-specific analysis we have carried out additional experiments to verify colocalization on Pyr cell soma in new Figure 3. Gephyrin.FingR and gephyrin-IF remain highly correlated in this analysis. 

3. In the same vein, is it possible to do Gephyrin IF on materials shown in Fig. 4 &5? In these two main figures of this paper, alignment of Gephn.FingR and PV-Syn was assessed, and the authors showed a substantial PV-Syn boutons without apposing Gephn.FingR. If immunolabeling of Gephyrin antibody can also substantiate this finding, it would provide even stronger support.

This is a good suggestion. However, due to technical limitations in experimental design we are not able to complete this experiment within the revision period. Notably, we have tried immunostaining using multiple other gephyrin and GABAAR subunit targeting antibodies (NeuroMab Gephyrin antibody 73-443; Synaptic Systems Gephyrin 147 011; Synaptic Systems Gephyrin Oyster650; Millipore GABAAR α1 06868; Millipore GABAAR γ2, AB5559), but only observed reliable immunofluorescence signal using Synaptic System’s directly conjugated antibody, gephyrin-oyster550. Since presynaptic PV-Cre x Synaptophysin-tdTomato and postsynaptic gephyrin-oyster550 conjugated antibody’s fluorescent proteins’ excitation and emission wavelengths overlap substantially, we were unable to perform this validation experiment.

We note that gephyrin intrabody validation for excitatory and inhibitory cortical neurons was previously published in Gross et al.’s 2013 Neuron paper, where they showed GPHN.FingR-GFP targeting of endogenous gephyrin using colocalization with gephyrin immunofluorescence.

4. Has it been done in WT cell culture systems of triple labeling of (1) PV interneuron axons, (2) presynaptic markers (either synaptic vesicle or active zone markers like bassoon or piccolo) and (3) gephyrin?  Could this provide another way to prove PV boutons without gephyrin signal?

We could not find evidence for quantitative characterization of triple labeled parvalbumin, bassoon, and gephyrin synapses in cortical cell cultures (especially on excitatory neuron somas). However, we note that inhibitory synapses have often been defined by the presence of gephyrin, so that gephyrin-negative sites might be excluded from synaptic analysis. In addition, low-density neuronal cultures often lack the complex array of inhibitory synapses that are present in brain tissue.

Another MAJOR concern is on section 2.4 “correlated light and EM….”

Fig. 6 is based on ONE WT animal perfusion-fixed with 2% glutaraldehyde plus 2% PF, which is a rather heavy fixation that may reduce the antigenicity of target proteins to be immunolabeled. If the labeling efficiency is compromised by this fixation, then only a fraction of the antigen would be labeled.  Lack of gephyrin labeling at some inhibitory synapses could simply be due to low labeling sensitivity.  The subtitle of Fig. 6 should be changed to “GABAergic synapses have lower gephyrin content on Pyr somas than on dendrites”.  In fact, this reviewer suggests that the entire section 2.4 be deleted because this line of evidence is weak, and the material used is totally different from the rest of the paper.

We agree with the reviewer’s concern and now present results from tissue prepared with a lighter fixation protocol and embedded in the hydrophilic resin LRWhite, which preserves antigenicity to a greater degree. We continue to observe inhibitory PV synapses without gephyrin immunolabel, even though other nearby inhibitory synapses within the same samples exhibit robust gephyrin immunolabel on multiple consecutive sections (up to 6 or 7 in some cases). We believe that these results, because they are obtained using a very different approach from the rest of the paper, provide important additional evidence for the lack of or the very low gephyrin content at a sizable proportion of PV inhibitory synapses on neuronal cell bodies. 

Additional MINOR point,

1. Figure panels are labeled with upper case, figure legends are in lower case.

Revised as suggested.

2. It was obvious in Fig. 3 that there is gephyrin punta unopposed by PV-Syn. The explanation appeared much later in line 194,195 that they may be linked to other non-PV interneurons. Perhaps this explanation could appear earlier in association with Fig. 3.

The following sentence has been moved earlier to line 191-192:

“Additionally, not all Gephyrin.FingR puncta were associated with presynaptic PV boutons (Figure 3C-F), suggesting that they might be linked to another somatically-innervating interneuron subtype, such as cholecystokinin (CCK) basket cells [20].”

3. 6E may not be a spine, could be a thin dendrite shaft. Has it ever been shown that a spine can receive both excitatory and inhibitory inputs?  The cytoplasmic content of this “spine” appears more like dendritic shaft.

Indeed, some dendritic spines in neocortex (about 5 - 10% in rat barrel cortex, for example) receive both an excitatory and an inhibitory synapse. These inhibitory synapses on spines are known for their plasticity in response to changes in sensory input as first shown by one of the authors (Micheva & Beaulieu, 1995, PNAS 92:11834), as well as subsequently by Knott et al, 2002, Neuron 34:265, and Villa et al, 2016, Neuron 90:662, among others. We classified the example in Figure 6E as synapse on spine using also the information from the adjacent sections in the series.

4. There are two “data not shown” (line 254, line 289) that could be either supplemented with additional info, e. g., (n) values in the text, or provided in supplementary data.

We appreciate the reviewer’s suggestion and have added two supplemental figures:

Figure S1: Fluorescence intensity of PV boutons and gephyrin puncta by alignment category.

Figure S2: Total gephyrin volume and median gephyrin puncta sizes are similar for L2/3 and L5 Pyr neu-rons.

Round 2

Reviewer 2 Report

The authors have answered adequately to all my questions.